# NDLINEAR:
# PRESERVING MULTI-DIMENSIONAL STRUCTURE FOR PARAMETER-EFFICIENT NEURAL NETWORKS

## ABSTRACT

In deep learning, processing multidimensional inputs (e.g., images, medical scans, and time series) is an important task that often requires flattening the inputs. We introduce *NdLinear*, a drop-in replacement for linear layers that operates directly on tensors, requiring no flattening. By applying transformations separately along each dimension, NdLinear preserves native data structure while achieving dramatic parameter reductions, often by orders of magnitude, with minimal memory overhead. We prove NdLinear maintains expressivity through structured Tucker decomposition while preserving VC-dimension scaling. Extensive experiments demonstrate NdLinear's capacity to achieve significant parameter reductions with substantial wall-clock efficiency gains and minimal memory overhead. For instance, our *NdLinear-LoRA* matches or exceeds standard LoRA on language reasoning tasks using up to $9\times$ fewer parameters. Experiments across CNNs, RNNs, Transformers, and MLPs on vision, language, time-series, and tabular tasks consistently demonstrate NdLinear's efficiency gains. While excelling at axis-separable tasks, NdLinear has limitations with entangled spatial interactions. By processing data in its original N-dimensional form, NdLinear provides a theoretically grounded, practical component for building more efficient neural architectures.

## 1 INTRODUCTION

Deep learning excels at processing multidimensional data, such as medical scans, videos and sensor arrays. Yet, a fundamental inefficiency persists throughout modern architectures. Consider what happens when a 32×32×32 voxel tensor reaches a linear layer: it gets flattened into a 32,768-dimensional vector, scrambling the careful spatial organization that earlier layers worked to extract. This pattern repeats everywhere linear layers appear: CNN classification heads flatten spatial feature maps, Transformer MLPs flatten structured token representations, and RNN projections flatten temporal features. Beyond this massive inefficiency, flattening forces networks to waste capacity rediscovering that adjacent voxels are related, that temporal continuity matters, and that feature dimensions have distinct meanings. While convolutions preserve local structure, they cannot replace linear layers for global reasoning, channel mixing, or final predictions. This leaves a critical gap: no existing layer can perform general linear transformations while respecting multidimensional structure.

To address this, we introduce **NdLinear**, a novel linear layer that operates natively on N-D tensors. Unlike vanilla linear layers, NdLinear processes data in its original N-D form by applying distinct linear transformations sequentially along each tensor dimension. This dimension-wise approach inherently preserves the data's N-D structure.

Preserving data's native organization allows NdLinear to offer compelling advantages: It enhances representational power by maintaining structural integrity and facilitating a more natural information flow. It dramatically reduces parameter counts (often by orders of magnitude vs. flattened layers) and ensures efficient computation without sacrificing performance. These efficiencies lead to faster training/inference and lower memory usage. As a general-purpose layer, NdLinear is a versatile, drop-in replacement for vanilla linear layers in many architectures, benefiting diverse domains by enabling models to better leverage N-D data characteristics.

Our work makes the following primary contributions, substantiated by extensive empirical validation:

- We articulate the critical limitations of flattening in conventional linear layers for N-D data.
- We introduce and formulate **NdLinear**, a novel N-D linear layer using sequential, dimension-wise transformations to preserve data structure and achieve substantial parameter efficiency.
- We empirically demonstrate through comprehensive evaluations that NdLinear significantly enhances model performance, or matches it with drastically fewer parameters, when replacing vanilla linear layers across diverse architectures (CNNs, RNNs, Transformers, MLPs) and data domains, including vision, language, time-series, and tabular data.
- We analyze the significant parameter and computational efficiency gains originating from NdLinear's dimension-wise transformation principle.

By demonstrating that preserving N-D structure yields both theoretical and practical advantages, NdLinear challenges the conventional wisdom that flattening is a necessary evil in neural architectures.

## 2 RELATED WORK

Neural networks process multidimensional data through three main approaches, each with fundamental limitations:

**Flattening-based Methods.** Standard linear layers reshape N-D tensors to 2D matrices, destroying spatial and dimensional relationships. This forces networks to relearn structure from scrambled features, requiring excessive parameters. For example, a $32 \times 32 \times 32$ tensor needs $\sim 10^9$ parameters when flattened, versus $\sim 10^3$ if structure were preserved.

**Specialized Architectures.** Convolution layers excel at 2D/3D spatial patterns but become unwieldy beyond three dimensions. Depthwise separable convolutions (Chollet, 2017a) and axial attention (Ho et al., 2019) reduce complexity through factorization but remain tied to specific spatial operations rather than general tensor transformations.

**Tensor Decomposition Layers.** Recent work applies tensor algebra to neural networks. Tensor Contraction Layers (TCL) (Kossaifi et al., 2020) use multilinear maps to compress activations to lower dimensions. Tensor Regression Layers (TRL) (Kossaifi et al., 2020) parameterize predictions using Tucker or Tensor-Train formats. Tensor-Train layers (Novikov et al., 2015) decompose weights for compression. While efficient, these methods focus on specific tasks, dimensionality reduction or regression, rather than general-purpose transformation.

**Gap in Current Methods.** Despite extensive work on structured layers, no method provides a true drop-in replacement for linear layers that: (1) operates directly on N-D tensors without flattening, (2) supports flexible dimension-wise transformation (expansion or compression), (3) integrates seamlessly with modern architectures (bias, normalization, dropout). This gap motivates NdLinear, which we introduce in Section 3.

We provide detailed mathematical comparisons between NdLinear and existing tensor methods (TCL, TRL, TT decomposition) in Appendix A, demonstrating how our approach differs fundamentally in design philosophy and implementation, leading to the superior empirical results shown in Section 4.

## 3 LINEAR TRANSFORMATION PRESERVING N-DIMENSIONAL INFORMATION

Vanilla linear layers cannot process input tensors directly. They require transforming the inputs into 2D matrices, destroying the original N-D structure. For an N-D input tensor $X \in \mathbb{R}^{B \times D_1 \times \cdots \times D_N}$ with batch size $B$ and feature dimensions $(D_1, \ldots, D_N)$, standard layers flatten it to $\mathbb{R}^{B \times \prod_i D_i}$ before applying a linear transformation. NdLinear suggests an alternative approach that transforms $X$ directly to $Y \in \mathbb{R}^{B \times H_1 \times \cdots \times H_N}$ without flattening, where each dimension $D_k$ maps to $H_k$, while preserving the N-D structure throughout the transformation.

The NdLinear transformation processes N-D tensors by applying separate linear transformations along each of their feature dimensions sequentially. This contrasts with vanilla linear layers, which flatten the N-D tensor into a 2D matrix, losing the original multidimensional structure. By operating

---

**Algorithm 1** NdLinear Transformation

---

**Require:** Input tensor $X \in \mathbb{R}^{B \times D_1 \times \cdots \times D_N}$ (batch size $B$, original feature dimensions $D_1, \ldots, D_N$),
  Target output feature dimensions $H_1, \ldots, H_N$,
  Learnable weight matrices $W_k \in \mathbb{R}^{D_k \times H_k}$ for $k = 1, \ldots, N$,
  Optional learnable bias vectors $b_k \in \mathbb{R}^{H_k}$ for $k = 1, \ldots, N$.
  Let $X_{\text{out}} \leftarrow X$.
  **for** $k = 1$ to $N$ **do**
    Let current shape of $X_{\text{out}}$ be $(B, S_1, \ldots, S_N)$, where $S_j = H_j$ for $j < k$, and $S_j = D_j$ else.
    The $k$-th feature dimension (original size $D_k$, current size $S_k = D_k$) is targeted for transformation.
    Permute $X_{\text{out}}$ to move its $k$-th feature dimension to the last position. Shape becomes $(B, S_1, \ldots, S_{k-1}, S_{k+1}, \ldots, S_N, D_k)$.
    Reshape $X_{\text{out}}$ to a 2D matrix $X_{\text{matrix}}$ of shape $\left( B \cdot \prod_{j \neq k} S_j, D_k \right)$.
    Apply linear transformation: $Y_{\text{matrix}} \leftarrow X_{\text{matrix}} W_k + b_k$. ($Y_{\text{matrix}}$ has shape $\left( B \cdot \prod_{j \neq k} S_j, H_k \right)$).
    Reshape $Y_{\text{matrix}}$ back to N-D form: $(B, S_1, \ldots, S_{k-1}, S_{k+1}, \ldots, S_N, H_k)$.
    Permute dimensions to place $H_k$ (the new size of the $k$-th feature dimension) back into the $k$-th feature position. Shape becomes $(B, S_1, \ldots, S_{k-1}, H_k, S_{k+1}, \ldots, S_N)$.
    Update $X_{\text{out}}$ with the result of this transformation.
  **end for**
  **return** $X_{\text{out}}$, now of shape $\mathbb{R}^{B \times H_1 \times \cdots \times H_N}$.

---

on each dimension independently while preserving the others, NdLinear retains the inherent structural information of the data throughout the transformation.

Conceptually, for an input tensor $X \in \mathbb{R}^{B \times D_1 \times \cdots \times D_N}$, NdLinear learns $N$ distinct weight matrices $W_1, \ldots, W_N$, where each $W_k \in \mathbb{R}^{D_k \times H_k}$ transforms the $k$-th dimension of the input from its original size $D_k$ to a new size $H_k$, with optional bias vectors $b_k \in \mathbb{R}^{H_k}$ per dimension. The transformation is applied iteratively: the output from transforming dimension $k$ becomes the input for transforming dimension $k + 1$. The procedure is detailed in Algorithm 1.

In practice, these operations (transposing, reshaping, matrix multiplication, then inverse reshaping and transposing) can be efficiently implemented using standard tensor library functions like `torch.tensordot` or `einsum`. The key is that each weight matrix $W_k$ only transforms dimension $D_k$ to $H_k$. This operation modifies all entries along the $k$-th mode, performing the same linear transformation on each mode-$k$ fiber of the tensor.

This sequential application can be expressed using tensor notation as a series of mode-$k$ tensor-matrix products (Kolda and Bader, 2009): $Y = X \times_1 W_1 \times_2 W_2 \cdots \times_N W_N$, where each product $X \times_k W_k$ transforms the $k$-th mode of the tensor using matrix $W_k$. (Biases $b_k$ are added after each mode-$k$ product). The intermediate result of $X \times_k W_k$ becomes the input for the $\times_{k+1} W_{k+1}$ product.

### 3.1 Preserving The Expressiveness of Vanilla Linear Layers

The dramatic parameter reduction of NdLinear raises a natural question: does this efficiency sacrifice model expressivity? We show that despite using fewer parameters, NdLinear maintains sufficient representational capacity.

**Theoretical Guarantee.** We analyze expressivity through VC-dimension, which measures a model's capacity to fit arbitrary patterns. Following Bartlett et al. (2019), networks with $P$ parameters have VC-dimension $\Theta(P \log P)$.

**Theorem 3.1** (Informal; see Appendix C for formal statement). *An NdLinear network with $P_{nd} = d(a + b + c)$ parameters for tensor dimensions $(a, b, c)$ and hidden dimension $d$ maintains VC-dimension $\Theta(P_{nd} \log P_{nd})$ as $d \to \infty$, matching the scaling of vanilla linear layers with $P_{std}$ parameters.*

**Practical Implication.** While NdLinear uses fewer parameters, the reduction is polynomial and not exponential in the VC-dimension bound. This theoretical guarantee is validated empirically: NdLinear often *improves* performance despite parameter reduction, suggesting the structured factorization acts as beneficial regularization rather than a limiting constraint.

**Empirical Evidence: Representation Compression.** To understand why fewer parameters improve performance, we measured von Neumann entropy of learned representations. On the Radius Bump task F.3 across varying difficulties, NdLinear consistently produces 15-30% lower entropy than parameter-matched dense networks while achieving equal or better test MSE (Appendix F.3). This lower entropy, indicating more compressed representations, aligns with recent theoretical work showing compressed representations generalize better (Skean et al., 2025). NdLinear's structured factorization inherently eliminates redundancy while preserving task-relevant information, explaining why dramatic parameter reduction enhances rather than hurts performance.

## 3.2 FEWER LEARNABLE PARAMETERS AND WALL-CLOCK SPEEDUPS

A key advantage of NdLinear is its parameter efficiency. Consider transforming $X \in \mathbb{R}^{B \times D_1 \times \cdots \times D_N}$ to $Y \in \mathbb{R}^{B \times H_1 \times \cdots \times H_N}$.

**Parameter Reduction.** Vanilla linear layers require $(\prod_i D_i) \times (\prod_i H_i)$ parameters, which grow exponentially with dimensionality. NdLinear requires only $\sum_{i=1}^{N}(D_i H_i)$ parameters, which grow linearly. For example, transforming a $32 \times 32 \times 32$ tensor to the same size requires $\sim 10^9$ parameters for vanilla linear but only $3{,}072$ for NdLinear, a reduction of six orders of magnitude.

**Computational Complexity.** NdLinear's FLOPs scale as $\mathcal{O}(B \cdot N \cdot D^{N+1})$ for cubic tensors with $D_i = H_i = D$, compared to $\mathcal{O}(B \cdot D^{2N})$ for vanilla linear layers. This yields order-of-magnitude speedups that increase with tensor dimensionality. The exact FLOP count is:

$$\text{FLOPs}_{\text{NdLinear}} = B \sum_{k=1}^{N} \left( \prod_{j<k} H_j \cdot \prod_{j>k} D_j \cdot D_k H_k \right) \tag{1}$$

**Memory Efficiency.** While theoretical analysis bounds peak memory overhead at $1/N$ of baseline, which is typically $\leq 33\%$ for 3D tensors (proof in Appendix C.2), empirical measurements show much smaller overhead in practice: only 1.1-2.0% increased peak memory across diverse architectures, with training time overhead below 1.6% (Section 4.5).

## 3.3 INDUCTIVE BIAS AND DOMAIN ALIGNMENT

NdLinear's efficiency stems from a strong inductive bias: it assumes dimensions can be transformed independently. This bias has clear benefits and limitations.

**The Separability Assumption.** NdLinear implements a rank-1 Tucker decomposition, assuming the transformation can be factorized across dimensions. Crucially, this structure persists through ReLU-like activations. For 1-homogeneous activations $\sigma$ (ReLU, GELU):

$$\sigma(T(x)) = \sum_{r=1}^{R} \bigotimes_{k=1}^{m} \sigma(u_r^{(k)}) \tag{2}$$

where output rank remains bounded even after nonlinearities. This ensures the separability bias persists through network depth rather than degrading.

**Domain Alignment.** NdLinear excels when data dimensions represent independent factors, such as spectrograms (frequency × time), sensor arrays (sensor × time), and tabular data, where features have separable effects. However, it struggles with dense cross-dimensional interactions like XOR patterns, checkerboards, or highly entangled spatial features where dimensions cannot be meaningfully separated.

## 3.4 QUANTIFYING THE TRADE-OFF: DIAL-A-BIAS EXPERIMENT

To precisely characterize when NdLinear's bias helps versus hurts, we designed a controlled experiment interpolating between separable and entangled patterns.

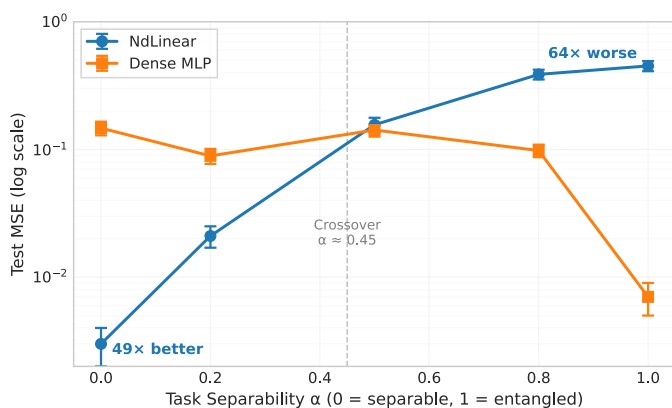

Figure 1: **NdLinear excels on separable tasks but struggles with entangled patterns** Performance comparison as task structure varies from purely separable ($\alpha = 0$) to fully entangled ($\alpha = 1$). The crossover at $\alpha \approx 0.45$ indicates NdLinear outperforms dense MLPs when tasks have $<45\%$ entanglement. This provides clear deployment guidance: use NdLinear for axis-aligned domains (spectrograms, time series, tabular data) but prefer dense layers for spatially entangled tasks (dense vision, XOR-like patterns).

We create synthetic targets that blend separable and entangled components:

$$y = (1 - \alpha) \cdot f_{\text{separable}}(X) + \alpha \cdot f_{\text{entangled}}(X), \quad \alpha \in [0, 1] \tag{3}$$

where $X \in [0, 1]^{32 \times 32}$ has i.i.d. uniform entries. The separable component $f_{\text{separable}}$ aggregates per-axis statistics (row/column means, variances), while $f_{\text{entangled}}$ requires cross-axis interactions (XOR patterns, checkerboards).

**Experimental Protocol.** We trained both NdLinear and dense MLPs (2 hidden layers, 256 units) on 10,000 samples for each $\alpha \in \{0, 0.1, ..., 1.0\}$.

**Results.** Figure 1 reveals a sharp crossover at $\alpha \approx 0.45$. NdLinear achieves 49× lower MSE for purely separable tasks ($\alpha = 0$) but 64× worse for fully entangled tasks ($\alpha = 1$). This provides clear guidance on the use of NdLinear, where the target structure aligns with axis-separable patterns.

## 4 EXPERIMENTAL RESULTS

We evaluate NdLinear as a drop-in replacement for vanilla linear layers across language, vision, time-series, and tabular domains. Our experiments span CNNs, RNNs, Transformers (including LLMs), MLPs, and DiTs, with model scales from 65K to 8B parameters on over 20 datasets. We also compare NdLinear against alternative structured layers including Tensor Regression Layers (TRL/TCL) and Tensor-Train (TT) decompositions (Novikov et al., 2015; Kossaifi et al., 2020)

**Universal findings:** NdLinear reduces parameters by 50-95% while maintaining or improving performance across all tested configurations. When compared to other structured approaches, Nd-Linear consistently achieves superior accuracy with lower computational overhead. We present domain-specific results below, with ablations in Section 4.5 and full details in Appendix E.

### 4.1 NATURAL LANGUAGE PROCESSING

#### 4.1.1 PARAMETER-EFFICIENT FINETUNING WITH NDLINEAR-LORA

We introduce NdLinear-LoRA, replacing LoRA's low-rank matrices $A, B$ with a single NdLinear adapter. Concretely, the LoRA update $\Delta W = BA$ is swapped for an NdLinear module that applies sequential mode-wise transforms on the reshaped activation tensor; all other LoRA mechanics (zero-init, scaling $\alpha$, residual addition, and merge-at-inference) remain unchanged. This preserves the native $N$-D structure during adaptation while keeping the same drop-in interface and training pipeline as standard LoRA.

Table 1: **NdLinear-LoRA demonstrates significant parameter efficiency for LLM finetuning, achieving comparable or improved accuracy over standard LoRA methods with up to $9\times$ fewer trainable parameters.** Accuracy ($\uparrow$) after LoRA of Qwen3-1.7B and LLaMA3-8B models.

| Model | Method | Params | Math | | CS Reasoning | | | |
|---|---|---|---|---|---|---|---|---|
| | | | GSM8K | M.Arith | CSQA | ARC-e | ARC-c | BoolQ |
| Qwen3-1.7B | LoRA ($r$=4) | 4.36M | 45.6 | 88.9 | 80.4 | 91.9 | **79.4** | 79.7 |
| | LoRA ($r$=8) | 8.72M | 40.3 | 82.2 | 80.9 | 91.8 | 79.3 | **80.8** |
| | NdLinear-LoRA | **1.15M** | **52.2** | **90.0** | **81.0** | **92.2** | 78.3 | 79.7 |
| LLaMA3-8B | LoRA ($r$=4) | 10.48M | 50.5 | **84.4** | 80.6 | 90.4 | 76.3 | **85.1** |
| | LoRA ($r$=8) | 20.97M | **51.6** | 81.1 | 81.7 | 89.0 | 73.6 | 76.5 |
| | NdLinear-LoRA | **2.26M** | 40.2 | 80.0 | **82.9** | **90.9** | **76.6** | 80.5 |

We fine-tuned Qwen3-1.7B (Yang et al., 2025) and LLaMA3-8B (Dubey et al., 2024) on mathematical reasoning (OpenMathInstruct-1 (Toshniwal et al., 2024)) and commonsense QA (CommonsenseQA training split (Talmor et al., 2019)), then evaluated on GSM8K, MultiArith, CSQA, ARC-Easy, ARC-Challenge, and BoolQ..

**Results.** Table 1 shows NdLinear-LoRA achieves superior performance with dramatically fewer parameters. On Qwen3-1.7B with 87% fewer parameters (1.15M vs 8.72M), it improves GSM8K by 11.9 points and MultiArith by 7.8 points over standard LoRA. On LLaMA3-8B with 9× fewer parameters (2.26M vs 20.97M), it achieves the best CSQA (82.9%) and ARC-Challenge (76.6%) scores. This suggests structured transformations capture task-specific patterns more efficiently than low-rank factorization.

### 4.1.2 LANGUAGE MODEL PRETRAINING

We pretrained OPT models (Zhang et al., 2022) (124M and 350M parameters) from scratch on BookCorpus and Wiki40B-English, replacing feedforward linear layers with NdLinear. Despite fewer parameters, NdLinear variants achieve lower perplexity (Table 5 in the Appendix). The improvement scales with model size: perplexity gap increases from 0.215 (OPT-Small) to 0.361 (OPT-Mid), suggesting NdLinear's benefits grow with scale.

### 4.1.3 TEXT CLASSIFICATION

We evaluated NdLinear in BERT's classification head (Devlin et al., 2019) on SST-2 sentiment analysis (Socher et al., 2013) and CoLA grammatical acceptability (Warstadt et al., 2018) tasks. We replaced BERT's standard two-layer classification head with a single NdLinear layer followed by a final linear projection.

**Results.** As shown in Table 7 in Appendix E, NdLinear improves both accuracy and ROC-AUC on both datasets while reducing the classification head parameters by approximately 85%. On SST-2, accuracy improves from 88.72% to 89.33%; on CoLA, from 77.90% to 79.06%.

## 4.2 TIME SERIES ANALYSIS

**Multivariate Time Series Forecasting.** We evaluated NdLinear in RNNs and Transformers for 12-hour ahead forecasting on four ETT (Electricity Transformer Temperature) datasets (Zhou et al., 2021), using 24 hours of historical input. For RNNs, we replaced vanilla linear layers with NdLinear. For Transformers, we replaced linear layers in feedforward blocks with NdLinear.

**Results.** Table 3 shows NdLinear consistently improves forecasting accuracy across all datasets while using 50% fewer parameters. For RNNs on ETTh1, MSE decreases from 0.290 to 0.088 (70% reduction). For Transformers on ETTh2, MSE decreases from 0.0226 to 0.0158 (30% reduction). These improvements demonstrate NdLinear's natural alignment with temporal-feature decomposition in multivariate time series.

**Time Series Classification.** We applied Transformer models with NdLinear to six UCR time series classification datasets (Dau et al., 2019), replacing vanilla linear layers in transformer blocks with NdLinear layers.

Table 2: **NdLinear significantly enhances Transformer-based time series classification, achieving superior F1 scores across UCR datasets with comparable or substantially fewer parameters than baseline models.** F1 Score and efficiency (# Params) of NdLinear compared to a standard Base Model and a parameter-comparable Base Model* (whose parameters are aligned with NdLinear's).

| Model | Metric | ECGFiveDay | HeartBeat | Chlorine Conc. | ECG5000 | LSST | Sleep |
|---|---|---|---|---|---|---|---|
| **Base Model** | Params | 3 363 | 4 323 | 12 900 | 12 966 | 29 343 | 12 966 |
| | F1 Score | 0.7668 | 0.7250 | 0.4436 | 0.8886 | 0.5928 | 0.4911 |
| **Base Model\*** | Params | 1 779 | 2 739 | 6 660 | 6 726 | 15 375 | 6 726 |
| | F1 Score | 0.7624 | 0.7214 | 0.4440 | 0.8878 | 0.5907 | 0.4897 |
| **NdLinear** | Params | 1 804 | 2 752 | 6 709 | 6 823 | 15 472 | 6 823 |
| | F1 Score | **0.7783** | **0.7363** | **0.4571** | **0.9058** | **0.6486** | **0.4978** |

Table 3: **NdLinear substantially improves time-series forecasting accuracy and dramatically reduces parameter counts in both RNN and Transformer models on Electricity Transformer Temperature (ETT) datasets.** With NdLinear, model parameters are reduced by 50%, alongside consistent gains in accuracy. Metrics reported are mean $\pm$ standard deviation over three experimental runs.

| Dataset | Method (Params: RNN / Trans.) | RNN | | Transformer | |
|---|---|---|---|---|---|
| | | MSE | MAE | MSE | MAE |
| ETTh1 | Linear (20.5k / 138k) | 0.2900 $\pm$ 0.0170 | 0.4060 $\pm$ 0.0246 | 0.0217 $\pm$ 0.0001 | 0.1158 $\pm$ 0.0004 |
| | NdLinear (9.6k / 70k) | **0.0880** $\pm$ 0.0115 | **0.2204** $\pm$ 0.0105 | **0.0173** $\pm$ 0.0003 | **0.0995** $\pm$ 0.0005 |
| ETTh2 | Linear (20.5k / 138k) | 0.2636 $\pm$ 0.0949 | 0.3955 $\pm$ 0.0748 | 0.0226 $\pm$ 0.0001 | 0.1229 $\pm$ 0.0003 |
| | NdLinear (9.6k / 70k) | **0.1536** $\pm$ 0.0137 | **0.2831** $\pm$ 0.0119 | **0.0158** $\pm$ 0.0019 | **0.0995** $\pm$ 0.0071 |
| ETTm1 | Linear (20.5k / 138k) | 0.0187 $\pm$ 0.0012 | 0.0926 $\pm$ 0.0039 | 0.0180 $\pm$ 0.0001 | 0.1005 $\pm$ 0.0005 |
| | NdLinear (9.6k / 70k) | **0.0174** $\pm$ 0.0017 | **0.0894** $\pm$ 0.0039 | **0.0161** $\pm$ 0.0006 | **0.0936** $\pm$ 0.0027 |
| ETTm2 | Linear (20.5k / 138k) | 0.0148 $\pm$ 0.0007 | 0.0825 $\pm$ 0.0047 | 0.0151 $\pm$ 0.0000 | 0.0965 $\pm$ 0.0001 |
| | NdLinear (9.6k / 70k) | **0.0139** $\pm$ 0.0009 | **0.0797** $\pm$ 0.0039 | **0.0141** $\pm$ 0.0001 | **0.0929** $\pm$ 0.0004 |

**Results.** Table 8 in Appendix E shows NdLinear reduces parameters by up to 47% while consistently improving F1 scores across all datasets. For example, on LSST, F1 improves from 0.5928 to 0.6486 with 47% fewer parameters. Even compared to parameter-matched baselines (Base Model*), NdLinear variants achieve superior performance, highlighting that the architectural advantage extends beyond mere parameter reduction.

### 4.3 TABULAR DATA

We evaluated NdLinear in MLPs on tabular classification (Cardio Disease dataset) (Sulianova, 2025) and regression (Food Delivery Time dataset) (Kumar, 2025). We replaced vanilla linear layers in two-layer feature extractors with NdLinear layers.

**Results.** Table 2 in shows NdLinear improves performance while dramatically reducing parameters. For Cardio Disease classification, NdLinear achieves higher accuracy (73.21% vs 72.65%) with 67% fewer parameters. For Food Delivery Time regression, NdLinear reduces MSE from 70.51 to 67.82 with 58% fewer parameters. These results demonstrate that NdLinear's dimensional factorization aligns well with tabular data where features often have independent effects.

### 4.4 COMPUTER VISION

#### 4.4.1 VISION TRANSFORMER DISTILLATION

We created student ViTs (NdViT) by replacing all linear layers in feedforward blocks with NdLinear, distilling from a pretrained ViT-B/16 teacher (Dosovitskiy et al., 2021). We tested various embedding dimensions (200, 300, 400) and depths (3, 6, 9 blocks) on CIFAR-10 and CIFAR-100 (Krizhevsky et al., 2009).

**Results.** Table 10 and Figure 3 in Appendix E show NdViT consistently outperforms standard ViT students while using 26-68% fewer parameters. Notably, NdViT with embedding dimension 200

surpasses a standard ViT with dimension 500, demonstrating superior parameter efficiency. The performance gap increases with model depth, suggesting NdLinear's benefits compound in deeper architectures.

### 4.4.2 CNN IMAGE CLASSIFICATION

We replaced the penultimate linear layer in CNNs with NdLinear on CIFAR-10 and CIFAR-100, and compared against Tensor Regression Layers (TRL/TCL) and Tensor-Train (TT) layers.

**Results.** NdLinear achieves higher accuracy with significantly fewer parameters on both datasets (Table 9 in Appendix E). On CIFAR-10, NdLinear improves accuracy by 2.6% while reducing parameters by 94%. On CIFAR-100, it improves accuracy by 5.1% with 60% fewer parameters.

**Comparison with Structured Methods.** Table 4 compares NdLinear against TRL/TCL and TT layers on CIFAR-100. NdLinear achieves superior accuracy (71.3%) compared to TRL/TCL (69.4%) and TT (56.2%), while using fewer parameters (434K vs 548K vs 769K), lower FLOPs (0.84G vs 3.97G vs 5.25G), 6× lower latency than TT (0.98ms vs 5.87ms), and minimal GPU memory overhead (35.16 MB vs 35.60 MB vs 100.44 MB). This demonstrates NdLinear's practical advantages over alternative structured approaches.

Table 4: **NdLinear outperforms existing structured tensor methods.** Comparison with Tensor Regression Layers (TRL/TCL) and Tensor-Train decomposition on CIFAR-100 classification, replacing the CNN's penultimate linear layer.

| Method | Mem (MB) | Acc@5 | Latency (s) | FLOPs (G) | Params |
|---|---|---|---|---|---|
| **NdLinear** | **35.16** | **0.7133** | **0.000976** | **0.843** | **433,588** |
| TRL/TCL | 35.60 | 0.6935 | 0.001116 | 3.97 | 548,032 |
| TT | 100.44 | 0.5617 | 0.005871 | 5.25 | 769,316 |

### 4.4.3 GENERATIVE MODELING WITH DIFFUSION TRANSFORMERS (DiT)

We evaluated NdLinear in DiTs through two approaches: training from scratch on ImageNet-100 (Russakovsky et al., 2015) and modifying pre-trained DiT models from (Jin and Xie, 2024). For training from scratch, we replaced linear layers in the timestep embedder. For pretrained models, we replaced linear layers in both timestep embedders and attention MLPs.

**Results.** When trained from scratch, NdLinear variants achieve lower (better) FID scores with comparable parameters (Figure 4 in Appendix E). For pretrained models, NdLinear reduces parameters from 674M to 619M while maintaining similar FID scores. demonstrating that parameter reduction doesn't sacrifice generation quality.

**Modifying Linear Layers in Pre-trained DiT Models** To evaluate performance maintenance with parameter reduction, we replaced standard Linear layers with NdLinear in pre-trained DiTs. This was done in both the timestep embedder and the MLP components of attention heads.

**Results.** Table 11 in Appendix E shows that NdLinear-based models (619M and 563M) achieve FID scores comparable to the larger baseline 674M model, despite significant parameter reductions.

### 4.5 ABLATION STUDIES

We conducted extensive ablations to understand NdLinear's design choices and practical considerations (full details in Appendix F).

### 4.5.1 DESIGN CHOICES

**Per-mode Bias Terms.** Including bias terms for each dimension transformation significantly improves performance, with benefits increasing at larger widths. On the Radius Bump task, bias terms improve MSE by 4.5% at width 32 and 15.2% at width 128, with negligible parameter overhead (<5%) (Table F.1 in the Appendix). This suggests per-mode biases help capture dimension-specific offsets. More information on the taks can be found in Appendix F.3

**Axis Ordering Robustness.** NdLinear shows remarkable robustness to transformation order. Permuting axes (original, reverse, random) cause only 4% accuracy variation on CIFAR-100, with random ordering achieving 96% of baseline performance (Table F.2 in the Appendix). This suggests NdLinear learns relatively axis-independent features, simplifying deployment.

**Hidden dimensions** offer a direct trade-off between efficiency and expressivity. Doubling hidden size improved accuracy by 2% while still using 40% fewer parameters than dense layers.

### 4.5.2 PRACTICAL CONSIDERATIONS

**Hyperparameter Robustness.** NdLinear requires no special tuning. Across learning rates (0.001, 0.01), batch sizes (32, 64, 128), and hidden configurations, accuracy varies only by 11 percentage points (64-75%) on CIAFR-10 (Table F.4 in the Appendix). Even the worst configuration (LR=0.01, batch=32) achieves 64% accuracy with 6× fewer parameters than dense layers, demonstrating robustness to suboptimal settings.

**Computational Overhead.** Despite sequential processing, measured overheads are minimal: peak memory increases by 1.1-2.0% (CIFAR CNN: 35.17→36.91 MB), training time by 0.6-1.6% (47.2→47.8s per epoch). Inference latency remains comparable due to reduced FLOPs offsetting sequential operations (Table F.6 in the Appendix).

**Sample Efficiency.** The separability bias dramatically affects data efficiency (Section F.5). On our synthetic tasks with $\alpha = 0.1$ separability, NdLinear reaches target error with 5× fewer samples (2,000 vs 10,000). Conversely, at $\alpha = 0.9$ entanglement, it requires 1.7× more samples (25,000 vs 15,000), quantifying the bias-variance trade-off. The representation analysis in Section 3.1 provides additional mechanistic insight into these efficiency gains.

**Architectural Comparisons.** As shown in Section 4.4.2, NdLinear outperforms alternative structured layers (TRL/TCL, TT) on all metrics, validating our design choices.

**Key Takeaway.** NdLinear is robust to implementation details and requires no special tuning, where standard hyperparameters work well. The critical decision is architectural: whether the task exhibits sufficient separability to benefit from NdLinear's bias.

## 5 CONCLUSION

We introduced NdLinear, a linear layer that operates directly on N-dimensional tensors through sequential dimension-wise transformations. This simple change, by processing tensors in their native form rather than flattening them, yields dramatic improvements across modern deep learning.

Our extensive experiments demonstrate that NdLinear reduces parameters by 50-95% while maintaining or improving performance across all tested configurations. The results reveal striking consistency: from NdLinear-LoRA fine-tuning LLaMA-8B with 9× fewer parameters while improving reasoning accuracy, to achieving 70% lower error in time-series forecasting with half the parameters, to outperforming alternative structured methods (TRL/TCL, TT). This universality, spanning language, vision, time-series, and tabular domains, validates dimension-wise factorization as a fundamental principle.

Theoretically, we proved NdLinear maintains expressivity through preserved VC-dimension scaling, explaining why dramatic parameter reduction doesn't sacrifice performance. Our controlled experiments precisely quantified when the method's inductive bias helps versus hurts: NdLinear excels when data exhibits axis-separable structure, but struggles with highly entangled patterns. This theoretical understanding, combined with our ablations showing robustness to hyperparameters and negligible computational overhead (<2%), provides clear deployment guidance.

Beyond its immediate practical benefits, NdLinear challenges a fundamental assumption in neural architecture design: that flattening is necessary for linear transformations. By demonstrating that structure-preserving operations consistently outperform their flattened counterparts, we reveal that the ubiquitous practice of tensor flattening has been systematically discarding valuable inductive biases. As neural networks evolve to process increasingly complex multidimensional data, from volumetric medical imaging to spatiotemporal climate models, NdLinear's principle of native dimensional processing offers not just an optimization, but a paradigm shift toward architectures that inherently respect and leverage the structure of our multidimensional world.

## ETHIC STATEMENT

This paper does not involve human subjects, personally identifiable data, or sensitive applications. We do not foresee direct ethical risks. We follow the ICLR Code of Ethics and affirm that all aspects of this research comply with the principles of fairness, transparency, and integrity.

## REPRODUCIBILITY STATEMENT

We ensure reproducibility of our experiments by fully describing the model architectures, datasets, preprocessing steps, hyperparameters, and training details in the main text and appendix. Code and scripts for reproducing the results are provided in the supplementary materials.

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
