## IMPACT STATEMENT

NdLinear replaces each dense linear layer with an $N$-mode rank-1 factorization, cutting parameters and FLOPs by up to two orders of magnitude on the benchmarks evaluated in this paper. The resulting drop in compute and memory lowers energy use and enables on-device inference and federated fine-tuning, broadening access to large-model capabilities for researchers and organizations with limited hardware. The authors will release full reference code to facilitate reproducible adoption. Because cheaper deployment also reduces the barrier for misuse, future work should evaluate privacy, robustness, and bias in models that adopt this compressed design.

## LIMITATIONS AND FUTURE DIRECTIONS

While NdLinear shows strong empirical efficiency, several promising research paths remain: (i) relax the rank-1 Tucker structure by permitting higher multilinear ranks or adding cross-mode residual connections to capture richer inter-mode interactions; (ii) develop memory-aware kernels that preserve compute and bandwidth efficiency as the number of modes $N$ or hidden size grows; (iii) learn or adapt the ordering of mode transforms to exploit data-driven structure; (iv) benchmark the layer in larger models ($> 10B$ parameters), new modalities such as 3-D medical imaging, streaming time-series, and edge devices, measuring accuracy, latency, and memory trade-offs; (v) derive approximation-error, sample-complexity, and optimization guarantees; and (vi) investigate privacy, robustness, and fairness when NdLinear enables lightweight federated deployment. Progress on these fronts will broaden NdLinear's applicability and deepen our understanding.

## LLM USAGE DISCLOSURE

We used large language models (LLMs) to aid and polish writing, such as improving clarity, grammar, and conciseness. We also used LLMs for retrieval and discovery, for example exhausting literature to identify potential missing related work. All technical content, proofs, experiments, and results are original contributions by the authors.

## A    DETAILED COMPARISON WITH RELATED METHODS

NdLinear's design as a structure-preserving, parameter-efficient linear layer for N-D tensors is informed by, yet distinct from, several established concepts in machine learning and tensor algebra. We detail these relationships below.

### A.1    TENSOR DECOMPOSITION METHODS

NdLinear's core mechanism employs mode-wise tensor-matrix products prominent in Tucker decomposition (Tucker, 1966; Kolda and Bader, 2009). However, unlike classical tensor decomposition methods that primarily analyze or compress static data or pre-existing weight tensors (Novikov et al., 2015; Newman et al., 2024), NdLinear integrates these mode-wise operations as a learnable, dynamic layer within a network. Its purpose is efficient forward transformation of activations while preserving N-D structure, not data analysis or post-hoc model compression.

NdLinear can be viewed as a *hand-crafted factorization* of a fully-connected weight matrix. Specifically, the full weight matrix $W_{\text{full}}$ implicitly has a Kronecker product structure derived from mode-wise matrices $\{W_1, \ldots, W_n\}$. This corresponds to a rank-1 Tucker decomposition without a core tensor (or equivalently, a core of rank 1 in each mode).

The main trade-off is expressiveness vs. efficiency. NdLinear's decomposition is low-rank in a multilinear sense—it cannot represent arbitrary non-factorizable interactions between dimensions. More flexible decompositions (full Tucker or higher-rank tensor decompositions) capture more interactions but require significantly more parameters compared to NdLinear's simple sum $\sum_i D_i H_i$ and can be more challenging to train, sometimes needing special initialization or multi-stage training.

If needed, one can extend NdLinear by increasing the factorization rank (e.g., learning multiple $W_i$ matrices per mode and summing their effects, analogous to a rank-$R$ core). However, our experiments

show the simple rank-1 version already performs well. Each $W_i$ clearly indicates how dimension $i$ is transformed, providing better interpretability than general tensor decomposition methods that disperse transformations across multiple factors.

## A.2 FACTORIZED CONVOLUTIONS AND AXIAL OPERATIONS

Neural nets have long exploited *axis-wise* structure to cut parameters and impose useful priors. In CNNs, an image tensor $X \in \mathbb{R}^{H \times W \times C}$ is *not* flattened; instead, convolutions slide local kernels over the two spatial axes $(H, W)$ while mixing channels $C$. In sequence models, a tensor $X \in \mathbb{R}^{L \times C}$ (length $L$, features $C$) is processed along the *sequence axis* $L$ (e.g., attention/conv) while mixing feature *channels* $C$ within positions. Below we recap common factorized operators and their limitations.

**Definitions.**  *Channels* are feature maps carried alongside a position-like axis (e.g., RGB planes or intermediate filters in CNNs; hidden features per token in sequences). *Sequence (or spatial) dimensions* are position-like axes along which locality or ordering matters (e.g., time $L$, image height $H$, width $W$).

**Grouped convolutions.**  Grouped convs restrict each filter to operate on a subset of channels, partitioning $C$ into groups to reduce parameters and FLOPs (Krizhevsky et al., 2012; Xie et al., 2017). *Limitation:* channel mixing is constrained within groups; cross-group interactions require additional layers, and the grouping choice is a manual architectural prior.

**Depthwise separable convolutions.**  Depthwise separable convs factor a full conv into a *depthwise* spatial conv per channel followed by a $1 \times 1$ (*pointwise*) conv that mixes channels (Chollet, 2017b; Howard et al., 2017). This yields large FLOP/parameter savings with strong accuracy. *Limitations:* the factorization is tied to 2D spatial structure; expressivity hinges on the pointwise mixer; extending beyond standard spatial axes typically needs custom kernels.

**Axial (factorized) attention.**  Axial attention decomposes 2D/3D attention into a sequence of 1D attentions applied along one axis at a time (e.g., height then width), cutting quadratic costs while preserving long-range interactions along each axis (Ho et al., 2019; Wang et al., 2020; Yan et al., 2023). *Limitations:* axis order becomes an architectural prior; full joint interactions across axes emerge only after stacking, and costs can still be high for very long axes.

**CNNs.**  CNNs are the canonical instance of *axis-aware* processing: they exploit spatial locality (factorization in $(H, W)$) and defer heavy channel mixing to $1 \times 1$ pointwise layers. Grouped/depthwise variants intensify this factorization to further reduce compute.

**NdLinear in this landscape.**  NdLinear generalizes the factorization principle beyond conv/attention mechanics: given an $n$-D activation $X \in \mathbb{R}^{D_1 \times \cdots \times D_n}$, it *systematically factorizes a linear map* into mode-wise transforms $D_i \to H_i$ and applies them *sequentially*, producing a structured output $Y \in \mathbb{R}^{H_1 \times \cdots \times H_n}$. Thus, instead of a single dense matrix $\mathbb{R}^{\Pi_i D_i} \to \mathbb{R}^{\Pi_i H_i}$, NdLinear uses $n$ small matrices with $\sum_i D_i H_i$ parameters, preserving the N-D shape throughout. Unlike grouped/depthwise convs (tied to spatial kernels) or axial attention (tied to attention mechanics and axis ordering), NdLinear: (i) applies to *arbitrary* N-D tensors (images, videos, spectrograms, multivariate time series, tabular tensors), (ii) naturally supports both *compression and expansion* per mode ($H_i \lessgtr D_i$), (iii) creates insertion points for normalization/activation between mode maps, and (iv) preserves a clear axis-wise inductive bias without requiring handcrafted groups or convolutional kernels.

**Known failure modes and how NdLinear relates.**  Axis-factorized operators can underperform when tasks require strong *entangled* cross-axis interactions (e.g., patterns tied jointly to $(H, W)$ rather than separably to each). Grouped/depthwise convs may also struggle if the chosen grouping misaligns with semantics; axial attention can be sensitive to axis order and depth. NdLinear shares the core trade-off (axis-wise separability vs. full expressivity) but makes it *explicit and tunable* by per-mode widths/ranks or stacking; it inherits the efficiency benefits of factorization while remaining N-D agnostic in form.

## A.3 Tensor Contraction and Regression Layers

**Tensor Contraction Layers (TCL).** TCLs (Cichocki, 2014; Novikov et al., 2015) implement a *one-shot* multilinear map by contracting an input tensor $X \in \mathbb{R}^{D_1 \times \cdots \times D_m}$ with a set of mode matrices $\{V^{(k)} \in \mathbb{R}^{R_k \times D_k}\}_{k=1}^m$, yielding a reduced tensor $X \times_1 V^{(1)} \times_2 \cdots \times_m V^{(m)} \in \mathbb{R}^{R_1 \times \cdots \times R_m}$. The goal is typically *dimensionality reduction* (feature compression) via tensor contractions prior to downstream layers; intermediate positions for normalization/activation and per-mode biases are usually not part of the basic formulation (Cichocki, 2014; Novikov et al., 2015).

**Tensor Regression Layers (TRL).** TRLs (Kossaifi et al., 2020) cast prediction as supervised regression/classification with a *fixed tensor format* for the weight (e.g., Tucker-/TT-structured). Given an input tensor $X$, a TRL fits a low-rank tensor $W$ (plus optional bias) such that $\langle W, X \rangle$ (or a nonlinear variant) matches targets. The emphasis is on *learning with low-rank weights* for sample/parameter efficiency; the tensor format (ranks/cores) is chosen a priori and does not expose interleaved normalization/activation between modes (Kossaifi et al., 2020).

**NdLinear vs. TCL/TRL.** NdLinear is a *structure-preserving, learnable linear layer* that applies *sequential* mode-wise maps to activations, $Y = X \times_1 W_1 \times_2 \cdots \times_m W_m \in \mathbb{R}^{H_1 \times \cdots \times H_m}$, with an implicit rank-1 Tucker/Kronecker weight and $\sum_k D_k H_k$ parameters. This design differs in intent and mechanics:

- **Goal.** TCL primarily *contracts* modes for reduction (Cichocki, 2014; Novikov et al., 2015); TRL fixes a low-rank *regression* format (Kossaifi et al., 2020); NdLinear is a *drop-in linear* alternative for N-D activations that preserves the full tensor shape and supports *expansion or compression* per mode ($H_k \gtrless D_k$).
- **Execution.** TCL performs a *single* contraction; NdLinear performs *sequential* per-mode maps, creating natural insertion points for LayerNorm/Dropout/activations and allowing *per-mode biases*. TRL optimizes a fixed low-rank weight but does not expose interleaved, mode-by-mode transforms during the forward pass.
- **Expressivity/control.** All three impose structured priors; NdLinear's axis-wise separability can be *tuned* via widths/ranks or stacked blocks, interpolating between strong separability and near-dense behavior while retaining N-D outputs.

**TCL as a special case of NdLinear.** Under the joint constraints *(i)* $H_k = R_k$ for all $k$ (no expansion beyond the contracted size), *(ii)* no per-mode biases, and *(iii)* no interleaved operations between mode maps (i.e., a single, commutative product), the one-shot TCL contraction coincides with NdLinear (Cichocki, 2014; Novikov et al., 2015). Outside this narrow corner, NdLinear is strictly more general and practical: it preserves N-D structure through sequential maps, supports per-mode biases and interleaving (stability), and flexibly expands or compresses each mode.

## A.4 Structured Linear Layers

NdLinear sits within the broader family of *structured* linear layers for parameter efficiency (Denil et al., 2013; Wei et al., 2024). Classic approaches constrain a large dense weight $W \in \mathbb{R}^{(\prod_i D_i) \times (\prod_i H_i)}$ by imposing algebraic structure *on $W$ itself*—e.g., low-rank factorizations (Sainath et al., 2013), block/Butterfly/Monarch-style sparse–fast transforms (Dao et al., 2022), or other learned structured matrices (Sindhwani et al., 2015; Potapczynski et al., 2024). These methods decouple the parameterization of $W$ from the native organization of the activations, often yielding strong compression but requiring the model to implicitly discover how that structure aligns with the data.

In contrast, NdLinear derives its structure *from the N-D layout of the input*. Given $X \in \mathbb{R}^{D_1 \times \cdots \times D_n}$, NdLinear replaces the dense map $\mathbb{R}^{\prod_i D_i} \to \mathbb{R}^{\prod_i H_i}$ with *sequential mode-wise transforms* $D_i \to H_i$, preserving tensor shape and inducing an explicit Kronecker (rank-1 Tucker) weight with only $\sum_i D_i H_i$ parameters. This data-centric factorization (i) makes the axis-wise inductive bias transparent and tunable (via per-mode widths/ranks or stacking), (ii) supports per-mode expansion or compression ($H_i \gtrless D_i$), and (iii) exposes natural insertion points for normalization/activation between modes—while retaining the efficiency benefits typical of structured linear layers (Denil et al., 2013; Wei et al., 2024; Sainath et al., 2013; Dao et al., 2022; Sindhwani et al., 2015; Potapczynski et al., 2024).

## A.5 GRAPH NEURAL NETWORKS

Graph Neural Networks (GNNs) address data with *irregular* connectivity by propagating information over edges via message passing (Scarselli et al., 2008; Micheli, 2009; Bronstein et al., 2017; Zhou et al., 2020; Wu et al., 2020). Concretely, a GNN updates node features by aggregating messages from neighbors and applying learnable transforms (Gilmer et al., 2017; Kipf and Welling, 2017; Hamilton et al., 2017). Stacking layers increases the receptive field and enables global interaction, but typically requires multiple rounds of propagation to mix distant nodes (Battaglia et al., 2018).

While an N-D grid (e.g., image or tensor lattice) *can* be modeled as a graph (one node per cell, edges to local neighbors), this introduces unnecessary overhead on regular grids: message passing is inherently local, so achieving global mixing along each axis often demands many layers, with added memory/compute and potential optimization issues (e.g., depth-related bottlenecks). Moreover, parameter sharing in GNNs is tied to edge types and neighborhood schemas, not directly to axis-wise tensor structure.

NdLinear takes the complementary route for *regular tensor grids*. Given $X \in \mathbb{R}^{D_1 \times \cdots \times D_n}$, it applies *global, mode-wise linear maps $D_i \to H_i$* in a single layer, mixing information *along entire axes* without constructing a graph or iterating local messages. This preserves the N-D layout, yields $\sum_i D_i H_i$ parameters via an implicit Kronecker (rank-1 Tucker) structure, and exposes insertion points for normalization/activation between mode maps. In short: GNNs excel when connectivity is irregular or non-Euclidean (Bronstein et al., 2017; Zhou et al., 2020), whereas NdLinear specializes in axis-aware, structure-preserving transformations on regular tensors, providing simpler and often faster global mixing along each dimension.

**When to use which.**   Use GNNs for arbitrary graphs, heterogeneous edge types, and relational reasoning on non-grid data (Scarselli et al., 2008; Micheli, 2009; Bronstein et al., 2017; Zhou et al., 2020; Wu et al., 2020). Use NdLinear when data are naturally arranged as regular tensors and you want efficient, axis-wise global interactions without flattening; its bias toward axis separability provides parameter savings and predictable behavior on grid-structured domains.

## A.6 OTHER SPECIALIZED APPROACHES

A variety of specialized architectures preserve structure without resorting to full flattening:

**Slicing-based layers.**   Methods that slice inputs along spatial/temporal (or rotated) subdomains process each slice with shared weights, then recombine (Shao et al., 2016; Dieleman et al., 2016). This preserves locality and orientation information with modest compute. *Limitations:* boundaries between slices can hinder cross-slice interaction; designs are task-/geometry-specific and often require bespoke preprocessing.

**Capsule Networks.**   Capsules use vector/matrix-valued units and routing to model part–whole hierarchies and pose relationships (Hinton et al., 2011; Sabour et al., 2017; Hinton et al., 2018). They maintain structured representations through learned agreement between capsules. *Limitations:* routing adds iterative, nontrivial overhead; scaling to large resolutions and datasets has proven challenging; design choices (routing, capsule size) are sensitive.

**Hadamard/Fourier feature mixing.**   Fixed orthogonal or Fourier-like transforms provide global mixing with $\mathcal{O}(N \log N)$ or even $\mathcal{O}(N)$ cost (e.g., Random Features, Fastfood, FNet, Block-based variants) (Rahimi and Recht, 2007; Le et al., 2013; Tancik et al., 2020; Lee-Thorp et al., 2022; Pan et al., 2022). *Limitations:* transforms are *fixed* (non-learnable) or only weakly parameterized, so alignment with data structure must be recovered by subsequent layers; expressivity depends on depth.

**Relation to NdLinear.**   NdLinear applies *learned*, factorized linear maps along each tensor mode, preserving the full N-D layout while enabling axis-wise global mixing with $\sum_i D_i H_i$ parameters. Unlike slicing (Shao et al., 2016; Dieleman et al., 2016), it does not require hand-crafted partitions; unlike capsules (Hinton et al., 2011; Sabour et al., 2017; Hinton et al., 2018), it avoids iterative routing; unlike fixed Hadamard/Fourier mixers (Rahimi and Recht, 2007; Le et al., 2013; Tancik et al., 2020; Lee-Thorp et al., 2022; Pan et al., 2022), it learns mode-wise transforms end-to-end. This

yields a simple, geometry-agnostic mechanism for structure-preserving, parameter-efficient linear transformation on regular tensors.

## B  More Related Work

Modern neural networks contain substantial parameter redundancy: a large fraction of weights can be predicted from a small subset, sometimes up to 95% with no loss in accuracy (Denil et al., 2013). This has motivated a broad line of work on *efficient parameterizations* that preserve accuracy while reducing storage and compute.

**Structured tensor factorization.**  A major thread leverages high-order structure via tensor decompositions of weights. CP/Tucker-style compressions applied to convolutional kernels reduce parameters and inference cost (Lebedev et al., 2015). Tensor Train (TT) layers compress fully-connected mappings into compact tensorized operators (Novikov et al., 2015). Block-Term (BT) tensor networks combine Tucker- and CP-like structure for additional flexibility (Ye et al., 2020). These tensor-structured layers reduce parameters while retaining rich multi-way interactions by factoring weights across modes.

**Structured matrices and parameter sharing.**  Another approach imposes algebraic structure on large dense matrices, replacing them with families that admit fast transforms and fewer degrees of freedom. Toeplitz-like and related structured operators provide strong compression with competitive accuracy (Sindhwani et al., 2015); related families (e.g., circulant, block-circulant) and low-rank factorizations likewise trade unrestricted expressivity for parameter/compute efficiency (Lebedev et al., 2015).

**Multi-space representations.**  Complementary to structural compression, multi-space learning embeds features into multiple geometries to better capture hierarchy and long-range relations. For example, jointly using Euclidean and hyperbolic spaces for LiDAR yields improved hierarchical encoding and pose estimation (Wang et al., 2023). Such representations enhance expressivity without necessarily increasing individual layer sizes.

**Preserving high-order structure in practice.**  Operational layers that respect native tensor axes often strike favorable accuracy–efficiency trade-offs. Depthwise separable convolutions split channel-wise and spatial mixing to cut FLOPs while preserving inductive bias (Chollet, 2017b; Howard et al., 2017). However, many fully-connected stages still flatten activations, discarding axis structure learned upstream.

**Positioning NdLinear.**  NdLinear aligns with these trends but differs in where structure is imposed: rather than factorizing *weights after flattening*, it performs *mode-wise* learned linear maps directly on N-D activations, preserving tensor layout throughout. Conceptually, it is a rank-1 Tucker (Kronecker) parameterization of the dense linear map, with parameters scaling as $\sum_i D_i H_i$ rather than $\prod_i D_i \prod_i H_i$. This data-aligned factorization complements tensorized weights (Lebedev et al., 2015; Novikov et al., 2015; Ye et al., 2020) and structured matrices (Sindhwani et al., 2015), and, like depthwise separable convolutions (Chollet, 2017b; Howard et al., 2017), leverages axis-aware inductive bias—without resorting to flattening.

## C  Proofs and Technical Details

### C.1  VC-Dimension Analysis

We analyze the expressive capacity of NdLinear compared to standard linear layers. Following Bartlett et al. (2019), any piecewise-linear feedforward network with $P$ parameters has VC-dimension $\Theta(P \log P)$.

**Theorem C.1** (VC-Dimension of NdLinear). *Consider input tensors of shape $(B, a, b, c)$ transformed to outputs of shape $(B, d, d, d)$. Let:*

- $N_{vanilla} = dabc$ *(parameters in vanilla linear layer)*

- $N_{nd} = d(a + b + c)$ *(parameters in NdLinear)*

*Then:*

1. *As $d \to \infty$ with $a, b, c$ fixed: $N_{nd} = \Theta(N_{vanilla})$*

2. *NdLinear's VC-dimension is $\Theta(N_{nd} \log N_{nd})$*

*Proof.* For part (1), observe that:

$$N_{nd} = d(a + b + c) = \frac{a + b + c}{abc} \cdot dabc = \frac{a + b + c}{abc} \cdot N_{vanilla} \tag{4}$$

Since $\frac{a+b+c}{abc}$ is a positive constant (for fixed $a, b, c$), we have:

$$N_{nd} = \Theta(N_{vanilla}) \quad \text{as } d \to \infty \tag{5}$$

For part (2), by the Bartlett et al. result, since NdLinear has $N_{nd}$ parameters and maintains piecewise-linear structure through ReLU activations:

$$\text{VCdim}_{NdLinear} = \Theta(N_{nd} \log N_{nd}) \tag{6}$$

$\square$

**Interpretation:** While NdLinear uses fewer parameters for finite $d$, as the hidden dimension grows, its parameter count becomes proportional to vanilla linear layers, preserving the same VC-dimension scaling.

**Theorem C.2** (Parameter Count Lower Bound). *For positive integers $a, b, c, d$:*

$$d(a + b + c) > \log(dabc) \tag{7}$$

*Proof.* We have:

$$\log(dabc) = \log d + \log a + \log b + \log c \tag{8}$$
$$\leq (d - 1) + (a - 1) + (b - 1) + (c - 1) \quad \text{(using } \log x \leq x - 1\text{)} \tag{9}$$
$$= d + a + b + c - 4 \tag{10}$$

Therefore:

$$d(a + b + c) - \log(dabc) \geq d(a + b + c) - (d + a + b + c - 4) \tag{11}$$
$$= (d - 1)(a + b + c) - d + 1 + 4 \tag{12}$$
$$= (d - 1)(a + b + c - 1) + 4 \tag{13}$$
$$\geq 4 > 0 \tag{14}$$

since $d \geq 1$ and $a + b + c \geq 3$ for non-trivial tensors. $\square$

## C.2 PEAK ACTIVATION MEMORY ANALYSIS

**Proposition C.1** (Peak Memory Overhead Bound). *For an input tensor with $m$ modes of sizes $(d_1, \ldots, d_m)$ where $\prod_{i=1}^{m} d_i = D$, and output sizes $k_i \leq d_i$, the peak additional activation memory for backpropagation satisfies:*

$$\frac{\text{extra activation memory}}{\text{baseline activation memory}} \leq \frac{\max_i(k_i/d_i) \cdot \min_i d_i}{m \cdot \min_i d_i} \leq \frac{1}{m} \tag{15}$$

*For typical 3D tensors (m=3), this overhead is at most 33%.*

*Proof.* During forward pass, NdLinear sequentially transforms each mode. For backpropagation, we store intermediate activations after each mode transformation.

After transforming $j$ modes, the tensor has shape:

$$(B, k_1, \ldots, k_j, d_{j+1}, \ldots, d_m) \tag{16}$$

The peak extra memory occurs at the stage with the largest intermediate tensor. Since $k_i \leq d_i$, each intermediate tensor has at most $BD$ elements. The baseline memory is also $BD$ elements.

In the worst case where all $k_i = d_i$, we have at most $(m - 1)$ intermediate tensors to store, but only one is needed at any given time during backprop (due to sequential processing). Therefore:

$$\text{Peak overhead} = \frac{BD \cdot \max_i(k_i/d_i)}{BD} \leq \frac{1}{m} \tag{17}$$

$\square$

## C.3 COMPUTATIONAL COMPLEXITY

**Proposition C.2** (Exact FLOP Count). *NdLinear transforming* $\mathcal{X} \in \mathbb{R}^{B \times D_1 \times \cdots \times D_N}$ *to* $\mathcal{Y} \in \mathbb{R}^{B \times H_1 \times \cdots \times H_N}$ *requires:*

$$FLOPs_{NdLinear} = 2B \sum_{k=1}^{N} \left[ \left( \prod_{j=1}^{k-1} H_j \right) \left( \prod_{j=k+1}^{N} D_j \right) D_k H_k \right] \tag{18}$$

*where the factor of 2 accounts for multiply-add operations.*

This is typically $\mathcal{O}(BND^{N+1})$ for $D_i = H_i = D$, compared to $\mathcal{O}(BD^{2N})$ for vanilla linear layers—yielding orders of magnitude savings as $N$ increases.

# D IMPLEMENTATION DETAILS AND TRAINING PROTOCOL

Here we present implementation details and training protocol of NdLinear (Algorithm 1). Implementing NdLinear in practice involves careful attention to efficiency and compatibility with existing deep learning frameworks. We outline key considerations in the following.

**Memory Efficiency.** Despite handling high-dimensional tensors, NdLinear is *memory-efficient* due to its factorized parameterization. The forward pass requires allocating intermediate tensors during each mode transformation (after each linear operation, the tensor has one updated dimension). However, these intermediate allocations are of the same order as the input/output size and significantly smaller than the memory required for a gigantic flattened weight matrix. Modern tensor libraries (PyTorch, TensorFlow) facilitate implementing transpose-reshape-multiply steps without excessive data copying. We ensure in-place operations where possible (e.g., using `view` in PyTorch). Operations primarily reuse the input buffer for output as each mode is transformed, ensuring modest peak memory usage.

**Parameter Initialization.** Each weight matrix $W_i$ can be initialized using standard strategies for linear layers (Xavier/Glorot (Glorot and Bengio, 2010) or Kaiming (He et al., 2015) initialization based on fan-in and fan-out). Since $W_i$ has fan-in $D_i$ and fan-out $H_i$, the initialization follows $\text{Uni}(-\sqrt{\frac{6}{D_i+H_i}}, \sqrt{\frac{6}{D_i+H_i}})$ for Xavier uniform, or analogous formulas for other initializations. This helps maintain stable gradients across modes. One subtle point is that if $n$ is large, each mode's weight is relatively small, mitigating the risk of extremely large fan-in. We observed no initialization-specific difficulties; indeed, NdLinear's parameter reduction may help avoid gradient explosion or vanishing issues in deep networks.

**Computational Overhead.** Factorized operations (multiple transpose and reshape operations with smaller matrix multiplications) are highly optimized in modern BLAS libraries. Practically, runtime is comparable to or faster than fully-connected layers with similar outputs, due to reduced total FLOPs. Python-level overhead is minimal; the algorithm can be implemented in a single forward function looping over modes. For moderate $n$ (up to 4 or 5 dimensions), this loop is short. Explicit loops or unrolling (NdLinear2d, NdLinear3d, etc.) are feasible, but a simple loop suffices. Autograd engines handle tensor operations seamlessly, allowing standard backpropagation. Each $W_i$ receives gradients normally from upstream gradients.

**Training Protocols.** NdLinear layers can be trained end-to-end with standard optimization algorithms (SGD, Adam) just like standard linear layers. Loss functions depend on the task (cross-entropy, MSE, etc.) and are unaffected by NdLinear. However, because NdLinear significantly reduces parameters compared to fully-connected layers, it tends to overfit less, possibly needing less aggressive regularization. Common techniques remain useful: *weight decay* (L2 regularization) on weights $W_i$, and optionally *dropout* between layers. Dropout can be applied before or after NdLinear; entries in the output tensor $Y$ can be dropped as usual. Specialized regularizers for factorized weights (norm regularization, orthogonality) may help further restrict solution spaces, though not required.

**Optimization and Convergence.** Practically, each $W_i$ is updated based on a portion of the overall error gradient (due to sequential mode transforms). In experiments, all $W_i$ matrices learned smoothly with default optimizer settings. If dimensionality varies significantly across modes, gradient clipping or adaptive learning rates per mode may be beneficial. Throughout our numerical investigations, training dynamics are stable overall — NdLinear layers integrated seamlessly into models without requiring special tuning. Standard protocols (learning rate schedules, early stopping criteria, etc.) used for equivalent models with dense layers apply here.

## E   FULL EXPERIMENTAL DETAILS AND RESULTS

We present full experimental details and more results on LoRA fine-tuning in appendix E.1; language-model pretraining (OPT and BERT) in appendix E.1; time-series prediction (RNN and Transformer) in appendix E.2; tabular data in appendix E.3; and vision tasks (CNN, ViT, and DiT) in appendix E.4. The complete experiment code is available at https://github.com/cyclone-trout/ndlinear_neurips.

### E.1   PARAMETER-EFFICIENT FINETUNING WITH LORA AND NDLINEAR-LORA

In our study, we utilized state-of-the-art transformer architectures to investigate the impact of targeted modu le adaptation. We selected two base models, Qwen3-1.7B-Base (Yang et al., 2025) and Meta-Llama-3-8B (Dubey et al., 2024), recognized for their robust performance across various tasks. To focus our adaptations, we targeted specific modules within these models, including `q_proj`, `k_proj`, `v_proj`, `o_proj`, `gate_proj`, `up_proj`, and `down_proj`. This approach allowed us to enhance model capacity and efficiency selectively.

For the adaptation process, we employed Low-Rank Adaptation (LoRA) techniques (Hu et al., 2022), specifically using both NdLinear LoRA and classic LoRA configurations. We explored a range of alpha values (1, 4, and 8) and rank settings (1, 4, and 8) to determine the optimal configuration for our models. This exploration was critical for understanding how different levels of parameter sharing and scaling affect model performance and generalization.

The training process was conducted using the AdamW optimizer, a choice informed by its effectiveness in managing the complexities of transformer models. We set the learning rate to $1 \times 10^{-4}$, which provided a suitable balance between convergence speed and training stability. The batch size was set to 1, a decision that facilitated the use of gradient accumulation to optimize GPU memory usage. To ensure the models could handle a wide variety of inputs, we set the maximum sequence length to 512 tokens. The models were trained over 2 epochs, a duration found to be sufficient for achieving significant performance improvements without excessive computational cost. To ensure reproducibility, we used a random seed of 42 across all experiments.

Our models were fine-tuned using two datasets: Math10K and CommonsenseQA. These datasets were chosen for their ability to challenge the models with both mathematical reasoning and common-sense understanding. For evaluation, we employed a diverse set of benchmark datasets, including GSM8K, MultiArith, ARC-C, ARC-E, and BoolQ. This selection allowed us to assess the models' generalization capabilities across different types of reasoning tasks.

The entire implementation was carried out on a single NVIDIA H100 GPU, using Hugging Face's `AutoModelForCausalLM` framework[1], integrated with our custom `NdLinear` adapter layer. Datasets were tokenized using the default tokenizer for each model, with padding applied to the `eos_token`. We employed label masking to exclude prompt tokens from loss computation, ensuring

---

[1] https://huggingface.co/docs/transformers/en/model_doc/auto

that training focused on the relevant portions of the input. Our implementation leveraged PyTorch, along with Hugging Face Transformers, PEFT, and Accelerate, to facilitate efficient model training and adaptation. Evaluation was performed in a zero-shot setting using greedy decoding, which provided a consistent measure of model performance without the variability introduced by sampling methods.

**Open Pretrained Transformer (OPT)** (Zhang et al., 2022). For OPT-Small, which originally contains 124M parameters, replacing the standard linear layers with NdLinear reduces the parameter count to 119M. Similarly, for OPT-Mid, the parameter count decreases from 350M to 337M after the substitution.

Table 5: Perplexity comparison for OPT-Small and OPT-Mid models with Linear vs. NdLinear layers.

|  | Linear | NdLinear |
|---|---|---|
| OPT-Small (Params) | 15.970 (124M) | **15.755** (**119M**) |
| OPT-Mid (Params) | 12.926 (350M) | **12.565** (**337M**) |

In Table 5, both the OPT-Small and OPT-Mid models achieve lower perplexity scores after replacing standard linear layers with NdLinear layers, despite having fewer parameters. Moreover, the performance improvement becomes more significant as model size increases, with the perplexity gap widening from $0.215$ in OPT-Small to $0.361$ in OPT-Mid. Figure 2 shows that OPT models with NdLinear feedforward layers achieve lower final training and evaluation losses compared to their counterparts using standard linear feedforward layers.

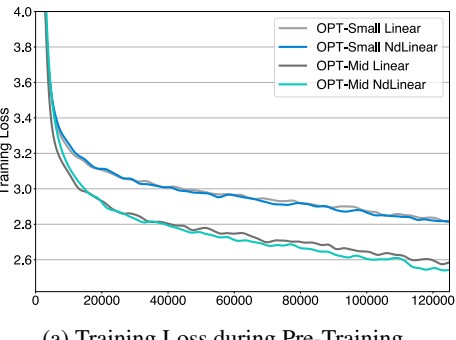
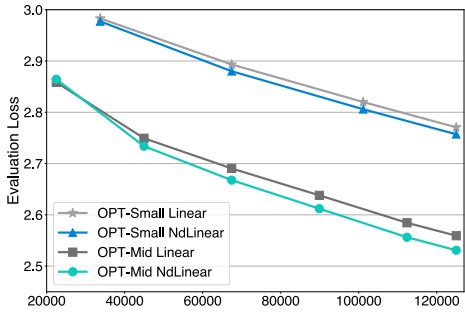

(a) Training Loss during Pre-Training.

(b) Evaluation Loss during Pre-Training.

Figure 2: Training and evaluation loss curves during OPT model pretraining. NdLinear variants consistently achieve lower loss values. x-axis represents the number of training steps.

*Zero-shot Tasks.* We also evaluate the OPTs' pretraining on 10 zero-shot NLP tasks:

- **Natural Language Inference Tasks**: CB (De Marneffe et al., 2019)
- **Coreference Resolution Tasks**: Winogrande (Sakaguchi et al., 2021)
- **Sentence Completion Tasks**: COPA (Roemmele et al., 2011), HellaSwag (Zellers et al., 2019)
- **Word Sense Disambiguation Tasks**: WiC (Pilehvar and Camacho-Collados, 2018)
- **Question Answering Tasks**: ARC-Easy, ARC-Challenge (Clark et al., 2018), OpenBookQA (Mihaylov et al., 2018), BoolQ (Clark et al., 2019)
- **Commonsense Reasoning Tasks**: PIQA (Bisk et al., 2020)

During evaluation, we cast all of the above tasks into a *multiple-choice format*. Namely, the goal is to select the correct completion from a set of candidate options. For each option, we compute the language model (LM) likelihood of the full input consisting of the context concatenated with the candidate completion. To account for differences in the lengths of candidate completions, we compute the average per-token log-likelihood for each option, following (Brown et al., 2020). The model's prediction is taken to be the option with the highest per-token likelihood.

**BERT** (Devlin et al., 2019). We replace the conventional two-layer linear classification head in BERT with an NdLinear layer followed by a classification layer. The NdLinear transforms have

Table 6: Perplexity Score and Zero-Shot Performance on OPT Model with and without NdLinear.

| | OPT-Small | | OPT-Mid | |
|---|---|---|---|---|
| | **Linear** | **NdLinear** | **Linear** | **NdLinear** |
| Num of Params | 124M | **119M** | 350M | **337M** |
| Perplexity | 15.970 | **15.755** | 12.926 | **12.565** |
| CB | 0.32 | **0.38** | 0.50 | **0.52** |
| Winogrande | 0.49 | **0.51** | 0.50 | 0.50 |
| COPA | **0.58** | 0.53 | 0.54 | **0.56** |
| HellaSwag | 0.26 | 0.26 | 0.28 | **0.29** |
| WiC | **0.51** | 0.50 | **0.50** | 0.49 |
| ARC-Easy | 0.29 | **0.30** | 0.29 | 0.29 |
| ARC-Challenge | 0.24 | **0.25** | **0.24** | 0.23 |
| OpenBookQA | 0.32 | **0.35** | 0.34 | 0.34 |
| BoolQ | 0.49 | **0.59** | **0.60** | 0.56 |
| PIQA | **0.53** | 0.50 | 0.53 | 0.53 |

hidden dimensions of $(2, 2)$. Each model is trained for 200 epochs with a batch size of 32, a hidden layer size of 128, and a learning rate of 0.005.

Table 7: BERT text classification performance on CoLA and SST-2 datasets. NdLinear improves accuracy and ROC AUC with $\approx 85\%$ fewer parameters in the classification head.

| Dataset | Method | Params (Head) | Accuracy | ROC AUC |
|---|---|---|---|---|
| CoLA | Linear | 1,544 | $0.7790 \pm 0.0143$ | $0.7127 \pm 0.0264$ |
| | NdLinear | 222 | $\mathbf{0.7906 \pm 0.0142}$ | $\mathbf{0.7405 \pm 0.0209}$ |
| SST-2 | Linear | 1,544 | $0.8872 \pm 0.0079$ | $0.8867 \pm 0.0080$ |
| | NdLinear | 222 | $\mathbf{0.8933 \pm 0.0093}$ | $\mathbf{0.8932 \pm 0.0073}$ |

## E.2 TIME SERIES

**Time Series Forecasting.** In our experiments using RNNs, we used a sequence length of 24 and a forecast horizon of 12 for all models. The models were trained for 100 epochs using the Adam optimizer with a learning rate of 0.02 and a batch size of 128. The dataset was split into training, validation, and evaluation sets with proportions of 60%, 20%, and 20%, respectively. We set the hidden size to 96, used a single recurrent layer, and applied a dropout rate of 0.3.

For Transformer-based Forecasting tasks, the experiments were conducted using a time series transformer model with a model dimension and hidden dimension both set to 32, a single transformer layer, and a dropout rate of 0.1. The GELU activation function was employed throughout, and the models were trained for 10 epochs with a batch size of 128 and a learning rate of 0.001. All models optimized using Adam and mean squared error as the loss function.

**Time Series Classification.** We set the hidden size of all RNN layers to 128 and used 3 recurrent layers, with a batch size of 32 and a learning rate of 0.005. Models were trained for 200 epochs using the Adam optimizer and cross-entropy loss.

### E.3 TABULAR DATA

We compare the performance of the Linear and NdLinear models. The Linear Model uses two linear layers for feature extraction, while the NdLinear Model replaces them with NdLinear layers.

**Classification.** Target labels were one-hot encoded. The Linear Model utilized fully connected layers with input dimension [11] and hidden dimension [128], followed by ReLU and a final linear output layer. The NdLinear Model used custom NdLinear layers with input dimensions [11, 1] and hidden dimensions [11, 64], also followed by ReLU and a final linear output layer. Both models are trained over 40 epochs with a batch size of 32 and a learning rate of 0.0001 using AdamW optimizer. Data was randomly shuffled and split into 80% training and 20% testing sets. Cross-entropy loss is used for training, and classification accuracy is used for model evaluation.

**Regression.** Target labels were kept as continuous values. The Linear Model utilized fully connected layers with input dimension [14] and hidden dimension [128], followed by ReLU and a final linear output layer. The NdLinear Model used custom NdLinear layers with input dimensions [14, 1] and hidden dimensions [32, 64], also followed by ReLU and a final linear output layer. Both models are trained over 40 epochs with a batch size of 32 and a learning rate of 0.0002 using AdamW optimizer. Data was randomly shuffled and split into 80% training and 20% testing sets. MSE loss is used for both training and model evaluation.

Table 8: NdLinear with MLPs on tabular datasets. For classification (Cardio Disease), the metric is Accuracy (higher is better). For regression (Delivery Time), the metric is MSE (lower is better).

| Dataset | Task | Method | #Params | Metric |
|---|---|---|---|---|
| Cardio Disease | Classif. (Accuracy) | Linear | 18 306 | 0.7265 |
| | | NdLinear | 5 962 | **0.7321** |
| Delivery Time | Regress. (MSE) | Linear | 18 561 | 70.508 |
| | | NdLinear | 7 873 | **67.824** |

### E.4 VISION

**Image Classification with CNN.** The NdLinear version uses three transforms with hidden dimensions of $(64, 8, 8)$, while the Linear version uses a single hidden dimension of 256. Models were trained for 50 epochs using Adam optimizer (learning rate 0.001), batch size 64, cross-entropy loss, and mixed-precision training on CUDA when available.

Table 9: Image classification with CNNs on CIFAR-10 (top-1 Acc.) and CIFAR-100 (top-5 Acc.). NdLinear achieves higher accuracy with fewer parameters.

| Dataset | #Params | Method | Accuracy |
|---|---|---|---|
| CIFAR-10 | 1.07M | Linear | $0.7426 \pm 0.0025$ |
| | 65k | NdLinear | $\mathbf{0.7689 \pm 0.0060}$ |
| CIFAR-100 | 1.09M | Linear | $0.6587 \pm 0.0075$ |
| | 433k | NdLinear | $\mathbf{0.7096 \pm 0.0121}$ |

**Vision Transformers (ViT) (Dosovitskiy et al., 2021).** Training used a batch size of 512, AdamW optimizer, learning rate $2.75 \times 10^{-4}$ for 30 epochs, and a distillation temperature of 3. Input images $(224 \times 224)$ were augmented with random cropping and horizontal flipping.

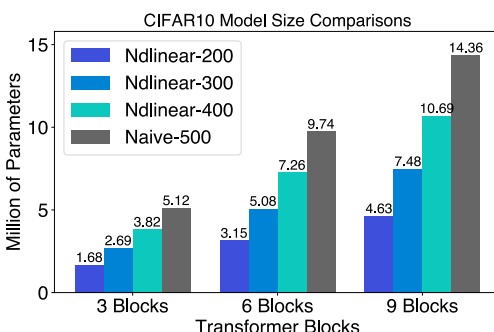 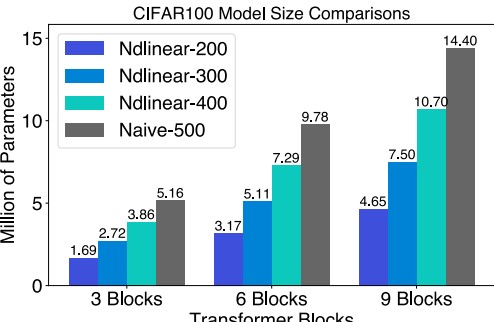

Figure 3: NdLinear's efficiency. Reduced ViT model parameter counts on CIFAR-10 and CIFAR-100 for a distillation task.

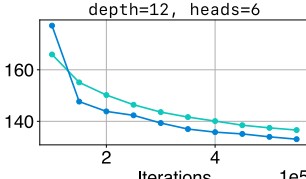 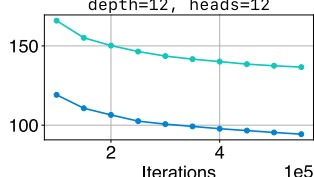 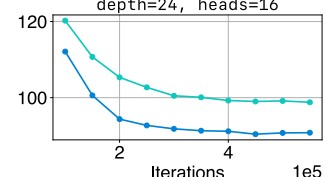

Figure 4: DiT achieving lower (better) FID scores for image generation on ImageNet-100 when trained from scratch with comparable parameters.

Table 10: NdViT vs. Naive ViT: Accuracy and parameter efficiency on CIFAR-10 (Acc@1) and CIFAR-100 (Acc@5). NdViT shows improved accuracy with fewer parameters.

| Dataset | Num. of Transformers | NdViT (Ours) | | | Naive |
|---------|---------------------|--------------|--------------|--------------|--------------|
| | | 200 | 300 | 400 | 500 |
| CIFAR10 | 3 Blocks | $65.77 \pm 0.47$ | $67.53 \pm 0.70$ | $\mathbf{69.00 \pm 1.27}$ | $62.09 \pm 0.40$ |
| | 6 Blocks | $68.48 \pm 0.75$ | $70.20 \pm 0.73$ | $\mathbf{72.03 \pm 0.46}$ | $65.19 \pm 0.64$ |
| | 9 Blocks | $70.27 \pm 0.35$ | $71.50 \pm 0.58$ | $\mathbf{72.53 \pm 0.54}$ | $68.52 \pm 1.24$ |
| CIFAR100 | 3 Blocks | $70.78 \pm 1.36$ | $73.10 \pm 1.06$ | $\mathbf{74.14 \pm 1.66}$ | $69.34 \pm 0.88$ |
| | 6 Blocks | $73.60 \pm 0.83$ | $75.07 \pm 0.14$ | $\mathbf{76.37 \pm 0.71}$ | $73.84 \pm 0.39$ |
| | 9 Blocks | $74.24 \pm 0.32$ | $75.52 \pm 0.73$ | $\mathbf{76.61 \pm 0.26}$ | $75.60 \pm 0.70$ |

**Diffusion Transformers (DiT)** (Peebles and Xie, 2023). We used a learning rate of $1 \times 10^{-4}$ for $256 \times 256$ images, mixed-precision (`bfloat16`) training, automatic gradient accumulation, and a batch size of 256, varying model depth and attention heads.

Table 11: FID-10k scores for DiT models: Pre-trained vs. NdLinear variants with fewer parameters.

| | NdLinear | NdLinear | Baseline |
|---|---|---|---|
| Parameter Count | 619M | 563M | 674M |
| FID-10k | 5.4876 | 5.9420 | 5.4109 |

# F   DETAILED ABLATION STUDIES

## F.1   PER-MODE BIAS IMPACT

We ablate per-mode bias terms across widths on the Radius Bump task.

We generate 1,000 train and 200 test samples. Hidden widths {16, 32, 64, 128}. We have NdLinear-MLPs replacing Linear blocks with NdLinear, using per-axis hidden shapes {(4,4), (8,8), (16,16), (32,32)} matched to the hidden widths above. For all models, training uses Adam (learning rate $10^{-3}$) for 4,000 epochs under identical schedules, with early stopping when training loss $< 10^{-4}$; loss is mean squared error (MSE). We report parameter count and test MSE (mean $\pm$ std).

The benefit of per-mode bias grows with width, reaching **+15.2%** MSE improvement at width 128.

Table 12: Per-mode bias ablation on Radius Bump

| Width | Bias | MSE (mean) | MSE (std) | Params |
|---|---|---|---|---|
| 16 | False | 0.00332264 | 0.00173532 | 36 |
| 16 | True | 0.00335201 | 0.00157289 | 46 |
| 32 | False | 0.00332793 | 0.00170700 | 72 |
| 32 | True | 0.00318228 | 0.00163146 | 90 |
| 64 | False | 0.00336938 | 0.00160766 | 144 |
| 64 | True | 0.00305152 | 0.00145630 | 178 |
| 128 | False | 0.00339010 | 0.00173152 | 288 |
| 128 | True | 0.00287480 | 0.00139560 | 354 |

## F.2   AXIS ORDERING SENSITIVITY

We permute axes (original, reverse, random) and measure CIFAR-100 performance retention. Finding: robustness to ordering ($\leq$ 4 pp spread).

Table 13: Axis ordering sensitivity (CIFAR-100)

| Variant (axes) | Accuracy vs. baseline |
|---|---|
| Original order | **100%** |
| Reverse | 99% $\pm$ 1% |
| Random | 96% $\pm$ 1% |

## F.3   RADIUS BUMP: LAST-LAYER ENTROPY, PERFORMANCE, AND COMPRESSION

We evaluate Dense MLP vs. NdLinear on the Radius Bump task across three difficulty levels (shell thickness $\sigma \in \{0.10, 0.20, 0.30\}$). Inputs are $x \in [-1, 1]^{10}$ with i.i.d. coordinates $x_i \sim U(-1, 1)$; the target is $y = \exp(-(\|x\| - 0.8)^2/(2\sigma^2))$. For each $\sigma$ we generate 1,000 train and 200 test samples. Architectures: (i) Dense MLP baselines with hidden widths {16, 32, 64, 128} and depths {2, 3}; (ii) NdLinear-MLPs replacing Linear blocks with NdLinear, using per-axis hidden shapes {(4,4), (8,8), (16,16), (32,32)} matched to the dense widths above and depths {2, 3}. For all models, training uses Adam (learning rate $10^{-3}$) for 4,000 epochs under identical schedules, with early stopping when training loss $< 10^{-4}$; loss is mean squared error (MSE). We report parameter count, forward-pass FLOPs (from analytical op counts), test MSE (mean $\pm$ std), and last-layer average output entropy (AveEntropy).

Across all $\sigma$, NdLinear variants consistently exhibit lower AveEntropy than parameter-matched dense models at similar widths/depths, while using markedly fewer parameters and FLOPs. At $\sigma = 0.20$ and $\sigma = 0.30$, depth-3 NdLinear achieves the *lowest* test errors (e.g., $0.00071 \pm 0.00019$ at width 64, and $0.00029 \pm 0.00021$ at width 64) together with *lower* entropies (0.344 and 0.341, respectively) than dense counterparts ($0.00128 \pm 0.00190$ with entropy 0.518; $0.00152 \pm 0.00029$ with entropy 0.521). For the hardest setting ($\sigma = 0.10$), NdLinear matches dense test error while maintaining

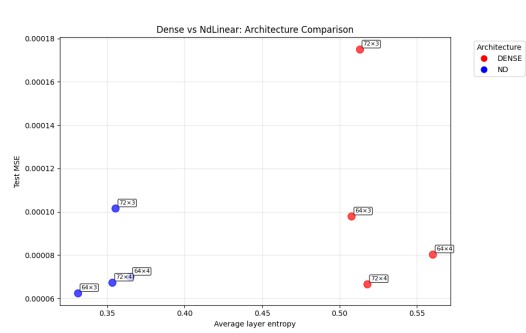

Figure 5: $\alpha = 0.1$ (hard): Narrow bump, very challenging.

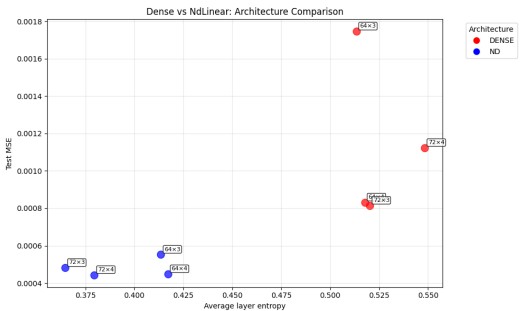

Figure 6: $\alpha = 0.2$ (medium): Moderate difficulty.

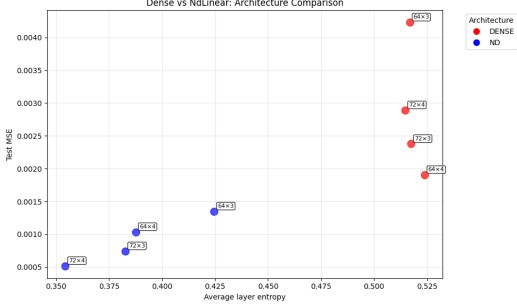

Figure 7: $\alpha = 0.3$ (easy): Wide bump, easier to approximate.

Table 14: Radius Bump ($\sigma = 0.10$; thin / hardest). Lower AveEntropy indicates more compact last-layer representations.

| Width | Kind | Depth | Hidden Shape | Params | Train MSE | Test MSE $\pm$ Std | FLOPs ($\cdot 10^9$) | AveEntropy |
|---|---|---|---|---|---|---|---|---|
| 16 | dense | 2 | (16) | 193 | 1.17e-5 | 0.00086 $\pm$ 0.00176 | 2.78 | 0.476 |
| 16 | dense | 3 | (16) | 465 | 1.37e-6 | 0.00087 $\pm$ 0.00189 | 6.70 | 0.625 |
| 16 | nd | 2 | (4, 4) | 46 | 1.19e-5 | 0.00086 $\pm$ 0.00186 | 0.66 | 0.405 |
| 16 | nd | 3 | (4, 4) | 86 | 1.18e-5 | 0.00086 $\pm$ 0.00186 | 1.24 | 0.518 |
| 32 | dense | 2 | (32) | 385 | 1.18e-5 | 0.00086 $\pm$ 0.00169 | 5.54 | 0.459 |
| 32 | dense | 3 | (32) | 1,441 | 2.93e-6 | 0.00084 $\pm$ 0.00188 | 20.75 | 0.560 |
| 32 | nd | 2 | (8, 8) | 90 | 1.20e-5 | 0.00086 $\pm$ 0.00182 | 1.30 | 0.321 |
| 32 | nd | 3 | (8, 8) | 234 | 1.16e-5 | 0.00086 $\pm$ 0.00178 | 3.37 | 0.426 |
| 64 | dense | 2 | (64) | 769 | 1.19e-5 | 0.00086 $\pm$ 0.00169 | 11.07 | 0.407 |
| 64 | dense | 3 | (64) | 4,929 | 1.59e-8 | 0.00085 $\pm$ 0.00196 | 70.98 | 0.507 |
| 64 | nd | 2 | (16, 16) | 178 | 1.12e-5 | 0.00086 $\pm$ 0.00175 | 2.56 | 0.292 |
| 64 | nd | 3 | (16, 16) | 722 | 1.12e-5 | 0.00086 $\pm$ 0.00180 | 10.40 | 0.377 |
| 128 | dense | 2 | (128) | 1,537 | 5.79e-6 | 0.00089 $\pm$ 0.00253 | 22.13 | 0.363 |
| 128 | dense | 3 | (128) | 18,049 | 1.84e-9 | 0.00089 $\pm$ 0.00158 | 259.91 | 0.469 |
| 128 | nd | 2 | (32, 32) | 354 | 1.12e-5 | 0.00086 $\pm$ 0.00158 | 5.10 | 0.295 |
| 128 | nd | 3 | (32, 32) | 2,466 | 8.71e-6 | 0.00086 $\pm$ 0.00204 | 35.51 | 0.286 |

Table 15: Radius Bump ($\sigma = 0.20$; medium). NdLinear often attains both lower error and lower last-layer entropy.

| Width | Kind | Depth | Hidden Shape | Params | Train MSE | Test MSE $\pm$ Std | FLOPs ($\cdot 10^9$) | AveEntropy |
|---|---|---|---|---|---|---|---|---|
| 16 | dense | 2 | (16) | 193 | 7.75e-4 | 0.00256 $\pm$ 0.00235 | 2.78 | 0.493 |
| 16 | dense | 3 | (16) | 465 | 1.29e-5 | 0.00166 $\pm$ 0.00118 | 6.70 | 0.592 |
| 16 | nd | 2 | (4, 4) | 46 | 9.91e-4 | 0.00273 $\pm$ 0.00273 | 0.66 | 0.410 |
| 16 | nd | 3 | (4, 4) | 86 | 1.01e-3 | 0.00275 $\pm$ 0.00272 | 1.24 | 0.521 |
| 32 | dense | 2 | (32) | 385 | 6.03e-4 | 0.00266 $\pm$ 0.00234 | 5.54 | 0.461 |
| 32 | dense | 3 | (32) | 1,441 | 5.92e-6 | 0.00180 $\pm$ 0.00117 | 20.75 | 0.560 |
| 32 | nd | 2 | (8, 8) | 90 | 9.92e-4 | 0.00273 $\pm$ 0.00232 | 1.30 | 0.323 |
| 32 | nd | 3 | (8, 8) | 234 | 1.19e-4 | 0.00192 $\pm$ 0.00089 | 3.37 | 0.428 |
| 64 | dense | 2 | (64) | 769 | 3.17e-4 | 0.00368 $\pm$ 0.00344 | 11.07 | 0.403 |
| 64 | dense | 3 | (64) | 4,929 | 1.84e-8 | 0.00128 $\pm$ 0.00190 | 70.98 | 0.518 |
| 64 | nd | 2 | (16, 16) | 178 | 8.35e-4 | 0.00252 $\pm$ 0.00216 | 2.56 | 0.293 |
| 64 | nd | 3 | (16, 16) | 722 | 9.56e-6 | 0.00071 $\pm$ 0.00019 | 10.40 | 0.344 |
| 128 | dense | 2 | (128) | 1,537 | 2.53e-5 | 0.00454 $\pm$ 0.00330 | 22.13 | 0.364 |
| 128 | dense | 3 | (128) | 18,049 | 2.01e-10 | 0.00159 $\pm$ 0.00102 | 259.91 | 0.471 |
| 128 | nd | 2 | (32, 32) | 354 | 6.07e-4 | 0.00233 $\pm$ 0.00211 | 5.10 | 0.296 |
| 128 | nd | 3 | (32, 32) | 2,466 | 2.33e-6 | 0.00081 $\pm$ 0.00019 | 35.51 | 0.284 |

*lower entropy* and large reductions in parameters and FLOPs. These patterns support the claim that lower last-layer entropy paired with equal-or-better error corresponds to better compression.

### F.4 HYPERPARAMETER SENSITIVITY

Grid search over learning rate, hidden size, and batch size on CIFAR-100. Finding: stable efficiency metrics and competitive accuracy across settings.

Note: GPU memory usage constant at 34.04-34.40 MB across all configurations.

### F.5 SAMPLE EFFICIENCY

Fix task structure and vary data size; measure samples needed to reach target error at different entanglement levels.

Table 16: Radius Bump ($\sigma = 0.30$; thick / easiest). NdLinear depth-3 variants achieve the lowest errors with the lowest entropies.

| Width | Kind | Depth | Hidden Shape | Params | Train MSE | Test MSE $\pm$ Std | FLOPs ($\cdot 10^9$) | AveEntropy |
|---|---|---|---|---|---|---|---|---|
| 16 | dense | 2 | (16) | 193 | 3.65e-3 | 0.00716 ± 0.00157 | 2.78 | 0.505 |
| 16 | dense | 3 | (16) | 465 | 1.87e-4 | 0.00210 ± 0.00084 | 6.70 | 0.627 |
| 16 | nd | 2 | (4, 4) | 46 | 5.91e-3 | 0.00827 ± 0.00261 | 0.66 | 0.406 |
| 16 | nd | 3 | (4, 4) | 86 | 2.73e-3 | 0.00710 ± 0.00345 | 1.24 | 0.527 |
| 32 | dense | 2 | (32) | 385 | 2.03e-3 | 0.00636 ± 0.00209 | 5.54 | 0.461 |
| 32 | dense | 3 | (32) | 1,441 | 1.42e-5 | 0.00166 ± 0.00063 | 20.75 | 0.588 |
| 32 | nd | 2 | (8, 8) | 90 | 3.53e-3 | 0.00564 ± 0.00190 | 1.30 | 0.323 |
| 32 | nd | 3 | (8, 8) | 234 | 6.78e-5 | 0.00045 ± 0.00021 | 3.37 | 0.424 |
| 64 | dense | 2 | (64) | 769 | 8.76e-4 | 0.00968 ± 0.00175 | 11.07 | 0.406 |
| 64 | dense | 3 | (64) | 4,929 | 1.34e-5 | 0.00152 ± 0.00029 | 70.98 | 0.521 |
| 64 | nd | 2 | (16, 16) | 178 | 3.01e-3 | 0.00502 ± 0.00142 | 2.56 | 0.293 |
| 64 | nd | 3 | (16, 16) | 722 | 9.46e-6 | 0.00029 ± 0.00021 | 10.40 | 0.341 |
| 128 | dense | 2 | (128) | 1,537 | 1.66e-4 | 0.01057 ± 0.00324 | 22.13 | 0.362 |
| 128 | dense | 3 | (128) | 18,049 | 2.03e-10 | 0.00249 ± 0.00051 | 259.91 | 0.474 |
| 128 | nd | 2 | (32, 32) | 354 | 1.78e-3 | 0.00460 ± 0.00096 | 5.10 | 0.296 |
| 128 | nd | 3 | (32, 32) | 2,466 | 3.41e-5 | 0.00041 ± 0.00014 | 35.51 | 0.278 |

Table 17: Hyperparameter sensitivity sweep (CIFAR-100)

| LR | Hidden Size | Batch | GFLOPs | Latency (s) | Params | Acc@5 (%) |
|---|---|---|---|---|---|---|
| 0.001 | 128,4,4 | 128 | 0.848 | 0.001415 | 232,876 | 72.75 |
| 0.001 | 256,2,2 | 64 | 0.901 | 0.001451 | 138,760 | 74.30 |
| 0.001 | 128,4,4 | 64 | 0.848 | 0.001381 | 232,876 | 72.67 |
| 0.001 | 128,4,4 | 32 | 0.848 | 0.001409 | 232,876 | 72.48 |
| 0.001 | 256,2,2 | 32 | 0.901 | 0.001388 | 138,760 | 73.80 |
| 0.001 | 256,2,2 | 128 | 0.901 | 0.001400 | 138,760 | **74.75** |
| 0.01 | 128,4,4 | 128 | 0.848 | 0.001318 | 232,876 | 71.62 |
| 0.01 | 128,4,4 | 64 | 0.848 | 0.001376 | 232,876 | 66.73 |
| 0.01 | 128,4,4 | 32 | 0.848 | 0.001364 | 232,876 | 66.56 |
| 0.01 | 256,2,2 | 64 | 0.901 | 0.001257 | 138,760 | 64.86 |
| 0.01 | 256,2,2 | 128 | 0.901 | 0.001312 | 138,760 | 70.20 |
| 0.01 | 256,2,2 | 32 | 0.901 | 0.001323 | 138,760 | 63.75 |

- **Nearly-separable tasks** ($\alpha = 0.1$): NdLinear reached target MSE with only **2,000** samples; parameter-matched linear model required **over 10,000** samples.
- **Highly-entangled tasks** ($\alpha = 0.9$): Standard linear model achieved target MSE with **15,000** samples; NdLinear struggled to match this performance even with **25,000** samples.

F.6 TRAINING AND MEMORY OVERHEAD

Measure peak activation memory and per-epoch training time after replacing a single dense GEMM with mode-wise GEMMs. Finding: empirical overheads are small ($< 3\%$ memory, $< 2\%$ time) as seen in Table 18.

Table 18: Empirical overheads across architectures

| Model | Peak Mem (MB) | Epoch Time (s) |
|---|---|---|
| CIFAR-100 CNN | 35.17 → 36.91 (+2.0%) | 47.2 → 47.8 (+0.6%) |
| ETTh1 RNN | 32.58 → 33.41 (+1.2%) | 12.3 → 12.6 (+1.2%) |
| Vision Transformer | 127.3 → 130.1 (+1.1%) | 179 → 185 (+1.6%) |

## F.7 COMPARISON WITH ALTERNATIVE STRUCTURED LAYERS

CNN head on CIFAR-100 comparing NdLinear to TRL/TCL and TT. Finding: NdLinear achieves higher Acc@5 with fewer params, lower FLOPs, and lower latency.

Table 19: NdLinear vs. TRL/TCL vs. TT (CIFAR-100 Acc@5)

| Method | Mem (MB) | Acc@5 | Latency (s) | FLOPs (G) | Params |
|---|---|---|---|---|---|
| **NdLinear** | **35.16** | **0.7133** | **0.000976** | **0.843** | **433,588** |
| TRL/TCL | 35.60 | 0.6935 | 0.001116 | 3.97 | 548,032 |
| TT | 100.44 | 0.5617 | 0.005871 | 5.25 | 769,316 |