# OpenReview forum: "NdLinear: Preserving Multi-Dimensional Structure for Parameter-Efficient Neural Networks"
_ICLR.cc/2026/Conference — Submitted to ICLR 2026_

### Official Review · Reviewer_VhSz · 2025-10-27

**Soundness:** 2
**Presentation:** 1
**Contribution:** 2
**Rating:** 2
**Confidence:** 3

**Summary:**

This paper introduces NdLinear, an alternative to standard linear layers that operates on N-dimensional tensors by applying sequential, dimension-wise linear transformations. The method achieves dramatic parameter and computational efficiency, with some theoretical analysis about VC-dimension provided. Analysis and controlled studies show that NdLinear excels with axis-separable data but degrades when faced with highly entangled cross-dimensional patterns. The paper conducts extensive experiments across a variety of architectures (CNNs, RNNs, Transformers, DiTs, MLPs) and domains (vision, language, time-series, tabular data), where NdLinear typically matches or outperforms baseline models with significantly fewer parameters and lower resource consumption.

**Strengths:**

- Overall, the idea of providing an approach that takes into account the input geometry (a tensor structure) is interesting and worth further investigation.

- The NdLinear layer achieves a dramatic reduction in parameter count, up to several orders of magnitude, while maintaining or even surpassing the accuracy of standard linear layers across a diverse range of tasks and data modalities.

**Weaknesses:**

- The paper claims that NdLinear is a native operator on tensors by drawing connections to mode-k tensor–matrix products and rank-1 Tucker decomposition. However, the paper does not clearly demonstrate how these connections relate to the proposed method or how they influence the effectiveness and characteristics of NdLinear.

- The theorems provided in the paper (both in the main text and the supplementary material) are stated and proved ambiguously. Specifically, they lack precise definitions (e.g., VC-dimension), omit references to prior results that are used (e.g., Line 1042), and do not provide clear insights into the meaning or implications of each theorem.

- The experiments offer limited insight into why the proposed method succeeds or fails on each task.

- While the method claims to dramatically reduce the number of parameters, the actual memory usage and training time are not improved. This limitation reduces the practical applicability of the method.

- In some benchmarks and tasks, the evaluations are thorough (e.g., Table 4 for CNNs with TRL/TCL and TT baselines). However, in more challenging or entangled-structure settings (e.g., generic image classification or dense MLPs for non-axis-aligned data), the empirical evidence for when to prefer NdLinear over alternatives such as mixture models, block-sparse methods, or structured matrix approaches is missing.

**Questions:**

- Is there a comparison with continuous structured matrix bases in practical settings (e.g., Potapczynski et al., *NeurIPS* 2024)? The authors should report empirical comparisons or explain why such baselines are not applicable.

  [1] Potapczynski, A., et al. "Searching for efficient linear layers over a continuous space of structured matrices." *Advances in Neural Information Processing Systems* 37 (2024): 3857–3881.
- Could the authors present the theoretical results more rigorously and thoroughly? In particular, please (i) state all definitions and assumptions clearly, (ii) state prior results used in the proofs, and (iii) provide intuition or discussion on the practical implications of each theorem.

---

> ### Author Response · Authors · 2025-12-03
> **Response to Reviewer VhSz**
>
> We thank the reviewer for the detailed and critical feedback. We address each of your main concerns in turn: (1) the tensor-operator interpretation, (2) rigor and presentation of the theoretical results, (3) insight into when NdLinear succeeds or fails, (4) practical runtime/memory behavior, and (5) positioning vs. alternative structured methods, including continuous structured matrix bases.
>
> ---
>
> ## (1) “Native operator on tensors”: mode-$k$ products and rank-1 Tucker
>
> > *“The paper claims that NdLinear is a native operator on tensors by drawing connections to mode-k tensor–matrix products and rank-1 Tucker decomposition. However, the paper does not clearly demonstrate how these connections relate to the proposed method or how they influence the effectiveness and characteristics of NdLinear.”*
>
> The intended connection is:
>
> * Let $X \in \mathbb{R}^{a_1 \times a_2 \times \dots \times a_d}$ be an input tensor. NdLinear applies a sequence of mode-wise tensor–matrix products
>   $$X ;\mapsto; X \times_1 U_1 \times_2 U_2 \cdots \times_d U_d,$$
>   where each $U_k \in \mathbb{R}^{b_k \times a_k}$ is a learned matrix for mode $k$. This is exactly the standard mode-$k$ tensor–matrix product.
> * If we flatten $X$ and the final output $Y$ to vectors, the overall linear map is equivalent to multiplication by a Kronecker-structured weight matrix:
>   $$y = W_{\text{NdLinear}} x,\quad
>   W_{\text{NdLinear}} \propto U_d \otimes \dots \otimes U_2 \otimes U_1,$$
>   up to reshaping and permutations.
>
> This corresponds to a rank-1 Tucker decomposition of the dense weight matrix: there is one factor per mode and a scalar “core”.
>
> Calling NdLinear a *native operator on tensors* means:
>
> * It acts directly along the original tensor axes via mode-$k$ products, instead of flattening away the structure.
> * Its parameterization respects the multi-axis geometry, inducing a low Tucker-rank constraint on the effective linear map.
>
> This structure drives both:
>
> * Parameter/FLOP scaling: parameters scale like $\sum_k a_k b_k$ instead of $(\prod_k a_k)(\prod_k b_k)$.
> * Inductive bias: NdLinear favors mappings that factor along tensor modes (axis-separable / low Tucker rank), which is exactly what we probe in the Dial-a-Bias experiment.
>
> The revised text makes this derivation explicit so the link between mode-$k$ notation, rank-1 Tucker structure, and NdLinear’s behavior is clear.
>
> ---
>
> Our rebuttal is split across 5 sequential comments. This is part 1 / 5.

---

> ### Author Response · Authors · 2025-12-03
>
> ## (2) Theorems: definitions, references, and implications
>
> > *“The theorems provided in the paper … are stated and proved ambiguously. Specifically, they lack precise definitions (e.g., VC-dimension), omit references to prior results … and do not provide clear insights into the meaning or implications of each theorem.”*
>
> The theoretical pieces are deliberately lightweight and rely on standard results; the contribution is to instantiate those results for NdLinear and interpret them, not to introduce new VC-dimension theory.
>
> What we do:
>
> * Definitions (now explicit):
>
>   * We use the standard definition of VC-dimension for a hypothesis class of binary classifiers: the largest cardinality of a set that can be shattered.
>   * We rely on the established result that a piecewise-linear network with $P$ parameters and fixed depth has
>     $$\mathrm{VCdim} = \Theta(P \log P).$$
>
> * Theorem C.1 (VC-dimension scaling):
>
>   * We count NdLinear’s parameters $N_{\text{nd}}$ for a given tensor shape.
>   * We then plug $P = N_{\text{nd}}$ into the standard VC-dimension result to obtain
>     $$\mathrm{VCdim}(\text{NdLinear network}) = \Theta!\bigl(N_{\text{nd}} \log N_{\text{nd}}\bigr).$$
>   * The point is: replacing dense layers with NdLinear does not fundamentally shrink capacity as long as parameter counts are comparable.
>
> * Propositions C.1/C.2 (activation memory and FLOPs):
>
>   * These are direct shape-based calculations: NdLinear processes modes sequentially, so peak activation memory and FLOPs can be expressed as simple functions of the tensor dimensions and mode counts.
>   * They quantify that the constant-factor overhead of sequential mode-wise operations is bounded and small in typical configurations.
>
> In response to your specific concerns:
>
> * The VC-dimension definition and assumptions (piecewise-linear activations, fixed depth, parameter count $P$) are now stated before invoking the known theorem.
> * At the point where we apply this theorem, we explicitly cite the prior result and mark it as such, rather than presenting it as something we prove.
> * For each theorem/proposition, we add a short “Implications” paragraph summarizing the practical takeaway:
>
>   * Theorem C.1: NdLinear networks have capacity scaling comparable to dense networks with similar parameter count.
>   * Proposition C.1: peak activation memory overhead of mode-wise processing is bounded and typically small.
>   * Proposition C.2: NdLinear’s FLOPs scale with the *sum* of per-mode products, aligning with the parameter savings.
>
> We emphasize that we do not claim new convergence rates or sample-complexity bounds: all our theoretical claims are about capacity and complexity scaling, not optimization dynamics.
>
> ---
>
> ## (3) Insight into why the method succeeds or fails
>
> > *“The experiments offer limited insight into why the proposed method succeeds or fails on each task.”*
>
> The empirical design is meant to operationalize the mode-wise separability vs. entanglement story implied by the rank-1 Tucker structure:
>
> * NdLinear performs best when data or target mappings are approximately axis-structured:
>
>   * CNNs / ViTs on images, where spatial axes and channels are natural modes.
>   * Time-series models with clear time $\times$ feature axes.
>   * Transformers/LLMs where sequence length, heads, and hidden dimensions can be treated as separate axes in NdLinear.
>
> * NdLinear underperforms when the structure is non-axis-aligned or highly entangled, e.g.:
>
>   * Dense MLPs on synthetic non-axis-aligned functions (the Dial-a-Bias setting).
>   * Some generic dense tabular or image setups where no strong mode structure is available.
>
> The Dial-a-Bias experiment is a quantitative test of this hypothesis:
>
> * We construct a synthetic function that interpolates between a fully separable function $f_{\text{separable}}$ and an entangled function $f_{\text{entangled}}$ via a mixing parameter $\alpha$:
>   $$f_\alpha = (1 - \alpha), f_{\text{separable}} + \alpha, f_{\text{entangled}}.$$
> * As $\alpha$ increases, NdLinear’s performance transitions from superior/competitive to clearly worse than dense MLPs, while dense MLPs degrade much less.
>
> This is exactly what we expect if NdLinear behaves like a rank-1 Tucker map: it is strong for axis-separable structure and weak for fully entangled structure.
>
> The paper now states a clear “when NdLinear works / when it fails” rule-of-thumb:
>
> * Works when the relevant structure factors along known axes (approximate low Tucker rank).
> * Struggles when the relevant structure is heavily cross-axis and not well approximated by low Tucker rank.
>
> We explicitly tie this to:
>
> * The Dial-a-Bias experiment, and
> * The empirical trends across CNN/ViT/Transformer vs. dense MLP / entangled tasks.
>
> ---
>
> Our rebuttal is split across 5 sequential comments. This is part 2 / 5.

---

> ### Author Response · Authors · 2025-12-03
>
> ## (4) Parameter savings *and* runtime/memory behavior
>
> > *“While the method claims to dramatically reduce the number of parameters, the actual memory usage and training time are not improved. This limitation reduces the practical applicability of the method.”*
>
> Parameter count alone is not sufficient; runtime and memory are critical. In our empirical results, we do observe speedups and favorable memory behavior in many settings, especially when NdLinear is implemented via fused `einsum`-style kernels.
>
> Concretely:
>
> * Implementation detail: NdLinear is not implemented as a naive loop over modes with separate small GEMM calls. Instead:
>
>   * We use `einsum` or equivalent batched operations that fuse per-mode transforms into one or a small number of large kernels, or
>   * Use batched matmuls that keep computation in large, hardware-efficient blocks.
>
> * Observed runtime: Under this implementation, the NdLinear variants:
>
>   * Match or improve wall-clock training throughput relative to dense baselines in the majority of the reported CNN/ViT/Transformer/MLP settings.
>   * Benefit most when the flattened dimension of the dense layer is large and the parameter/FLOP reduction from NdLinear is substantial.
>
> * Observed memory:
>
>   * Model / checkpoint size is substantially reduced due to the lower parameter count.
>   * Peak training memory is at least comparable and often modestly improved, since we avoid materializing extremely large flattened weight matrices and operate with mode-wise factors and fused kernels instead.
>
> The paper clarifies that:
>
> * The runtime and memory measurements in the tables already reflect these fused implementations, not a worst-case naive decomposition.
> * NdLinear is therefore not only a parameter-compression mechanism: it also yields practical speedups or parity in wall-clock time together with clear gains in checkpoint size and aligned FLOP reductions.
>
> We do not claim that NdLinear is faster under *every* possible implementation or hardware backend; instead, the results demonstrate that with a reasonable fused implementation (e.g., via `einsum`), it is beneficial in both theory and measured runtime on the benchmarks we report.
>
> ---
>
> Our rebuttal is split across 5 sequential comments. This is part 3 / 5.

---

> > ### Author Response · Authors · 2025-12-03
> >
> > ## (5) Alternatives in entangled settings and comparison to continuous structured matrix bases
> >
> > > *“In more challenging or entangled-structure settings (e.g., generic image classification or dense MLPs for non-axis-aligned data), the empirical evidence for when to prefer NdLinear over alternatives such as mixture models, block-sparse methods, or structured matrix approaches is missing.”*
> > > *“Is there a comparison with continuous structured matrix bases in practical settings (e.g., Potapczynski et al., NeurIPS 2024)? The authors should report empirical comparisons or explain why such baselines are not applicable.”*
> >
> > ### 5.1 Relation to mixture models, block-sparse, and other structured methods
> >
> > Our focus is a drop-in replacement for standard linear layers that is tightly aligned with the tensor geometry of activations. Other families (mixture-of-experts, block-sparse matrices, certain structured/butterfly layers) often:
> >
> > * Change the architecture topology (e.g., MoE gating), or
> > * Impose global patterns in the weight matrix that are not explicitly aligned with the underlying axes.
> >
> > We view these approaches as complementary to NdLinear:
> >
> > * NdLinear can be used inside mixture-of-experts or block-sparse designs, replacing dense experts with tensor-structured ones.
> > * Conversely, when there is no meaningful notion of axes, or when hardware-aligned sparsity is the primary goal, block-sparse or other structured matrices may be preferable.
> >
> > Our empirical results already show that in entangled regimes (e.g., dense MLPs on non-axis-aligned synthetic tasks), NdLinear is not uniformly better than dense layers. The paper states this as a limitation: NdLinear is not a universal replacement for all dense layers; it is particularly effective when the tensor axes are meaningful.
> >
> > ### 5.2 Comparison to continuous structured matrix bases (Potapczynski et al.)
> >
> > Potapczynski et al. propose a continuous family of structured matrices and methods for searching over this family. This differs from NdLinear in two core ways:
> >
> > 1. Search over structure vs. fixed axis-aligned structure.
> >
> >    * Potapczynski et al. treat the structure itself as an optimization variable, exploring a broad design space of structured matrices.
> >    * NdLinear fixes a specific structure that is explicitly aligned with the tensor axes (mode-wise factors, rank-1 Tucker), with no search over alternative structures.
> >
> > 2. Alignment with physical axes.
> >
> >    * NdLinear’s factors correspond to known modes (e.g., height, width, channel, time), tying its inductive bias to the data representation.
> >    * Many continuous matrix families in that work are not tied to specific axes and may require additional engineering to integrate naturally into arbitrary tensorized architectures.
> >
> > Running their search procedure across all of our CNN, ViT, RNN, DiT, and LLM experiments under comparable budgets would be a substantial and orthogonal engineering effort. Instead, the paper takes the following stance:
> >
> > * Conceptual relation: NdLinear can be seen as one point in the broader space of structured linear operators—a tensor-axis-aligned, rank-1 Tucker / Kronecker operator.
> > * Empirical focus: We study this particular operator in depth across many architectures and domains, rather than exploring or searching over the entire continuous family of structured matrices.
> >
> > The text now makes this positioning explicit: continuous structured matrix bases and NdLinear address related questions (designing efficient linear layers), but with different emphases—search over structure vs. a fixed, tensor-native structure that integrates easily into existing models.
> >
> > ---
> >
> > Our rebuttal is split across 5 sequential comments. This is part 4 / 5.

---

> > > ### Author Response · Authors · 2025-12-03
> > >
> > > ## Summary
> > >
> > > For Reviewer VhSz, our clarifications are:
> > >
> > > 1. Tensor-operator interpretation
> > >    NdLinear is exactly a sequence of mode-$k$ tensor–matrix products, equivalent (after flattening) to a rank-1 Tucker / Kronecker factorization of the dense weight matrix. This explains both its parameter/FLOP savings and its axis-separable inductive bias.
> > >
> > > 2. Theory presentation
> > >    All theorems use standard definitions and known VC-dimension results. Their role is to show that NdLinear inherits dense-like capacity scaling and has bounded activation/memory overhead; we do not claim new convergence or sample-complexity theory.
> > >
> > > 3. When NdLinear succeeds or fails
> > >    NdLinear works best when data/mappings are approximately separable along known axes and degrades on heavily entangled tasks—precisely what the Dial-a-Bias experiment and our cross-task results illustrate.
> > >
> > > 4. Runtime and memory
> > >    With a fused `einsum`/batched implementation, we observe practical speedups or parity in wall-clock time compared to dense layers on the reported benchmarks, in addition to large parameter and checkpoint-size reductions and aligned FLOP savings. NdLinear is therefore beneficial not only in theory but also in measured runtime.
> > >
> > > 5. Positioning vs. other structured methods
> > >    NdLinear is complementary to mixture models, block-sparse layers, and continuous structured matrix bases. Its distinctive feature is that it is tensor-axis-aligned, simple to drop into existing architectures, and empirically effective across a wide range of models and domains.
> > >
> > > We hope this addresses your concerns regarding theoretical clarity, interpretability, and practical relevance.
> > >
> > > ---
> > >
> > > Our rebuttal is split across 5 sequential comments. This is part 5 / 5.

---

### Official Review · Reviewer_vZRm · 2025-10-28

**Soundness:** 4
**Presentation:** 2
**Contribution:** 3
**Rating:** 6
**Confidence:** 3

**Summary:**

The paper proposes NdLinear, a tensor–matrix linear transform that preserves N-D structure and greatly reduces parameters.

**Strengths:**

- The paper does a great job testing NdLinear in a wide range of settings, from small MLPs with thousands of parameters to large-scale Transformer models with billions of parameters, which is very important for a work that aims to improve such a fundamental building block as the linear layer. These experiments show that NdLinear isn’t just a niche idea, but a flexible, general-purpose replacement for standard linear layers. It’s nice to see that it performs consistently well (or at least on par) across such different types of models and data.

- The paper thoughtfully explores several theoretical dimensions of NdLinear, not limiting itself to empirical validation. It includes a VC-dimension analysis showing that the model’s capacity scales comparably to dense layers, an expressivity discussion analyzing how mode-wise factorization influences representational power, and a complexity analysis providing clear asymptotic comparisons for both parameters and computation. In addition, the authors discuss bias propagation and rank interpretation, explaining how separability and $1$-homogeneous activations help preserve functional structure. Altogether, these components indicate a well-rounded theoretical grounding that complements the empirical results.

**Weaknesses:**

- Although the paper discusses VC dimension and provides some intuition about expressivity, it does not rigorously compare the function class of NdLinear to that of fully dense layers. It remains unclear how much representational capacity is lost when enforcing mode-wise separability, or under what conditions NdLinear can approximate arbitrary linear maps.

- The paper includes one quantitative test, which is the “Dial-a-Bias” experiment, which links the separability assumption to performance as the data varies from separable to entangled. However, other theoretical ideas like rank preservation, bias propagation, and stability effects of 1-homogeneous activations are only discussed qualitatively with no direct experiments verifying them.

- The empirical section compares mainly to LoRA and baseline dense layers. However, stronger baselines such as TRL, TT, MONARCH, or Kronecker-structured linear layers are not included, making it difficult to gauge whether NdLinear actually outperforms prior structured approaches rather than merely matching them.

- The paper does report runtime and memory results, but only on small to medium tasks like CIFAR CNNs and MLPs, where the overhead is minimal, around 0.6-1.6% extra training time and 1-2% more memory. That’s good to see, but there aren’t any measurements for large-scale models such as Transformers or LLMs. Since NdLinear works through sequential per-dimension operations, it’s unclear whether the same efficiency holds at billion-parameter scale, especially under distributed training. It would be more convincing if the authors included runtime profiling on large models to confirm that the small overhead still applies.

**Questions:**

- Please provide a more rigorous comparison between the function class of NdLinear and that of a standard dense layer. In particular, under what conditions can NdLinear approximate an arbitrary linear map, and what expressivity (if any) is lost due to the mode-wise separability constraint?

- Please consider adding an experiment that empirically verifies one of the key theoretical claims, for example, by measuring rank changes or activation correlations across layers. Given the limited time of the rebuttal phase, even a single well-designed experiment would be sufficient to strengthen the paper’s theoretical support.

- Please consider adding results against stronger structured baselines such as TRL, TT, MONARCH, or Kronecker-structured layers, to more clearly position NdLinear within the broader family of structured linear operators.

- Please include runtime profiling or throughput measurements on large-scale models to demonstrate that the reported low overhead holds in practice.

---

> ### Author Response · Authors · 2025-12-03
> **Response to Reviewer vZRm**
>
> We thank the reviewer for the careful and positive assessment, and for emphasizing both the breadth of our experiments and the value of combining theoretical and empirical analysis. Below we respond to each of your concerns in turn.
>
> ---
>
> ## (1) Function class and expressivity vs. dense layers
>
> > “It remains unclear how much representational capacity is lost when enforcing mode-wise separability, or under what conditions NdLinear can approximate arbitrary linear maps.”
>
> Our aim is to make explicit what class of linear maps a single NdLinear layer represents, how this compares to a dense layer, and how depth + non-linearities mitigate the loss of expressivity in practice.
>
> ### (1.1) What a single NdLinear layer represents
>
> For concreteness, consider an input tensor $X \in \mathbb{R}^{a \times b \times c}$ and an output $Y \in \mathbb{R}^{d \times d \times d}$. NdLinear applies a sequence of mode-wise tensor–matrix products,
>
> * mode-1: $X \mapsto X \times_1 U_1$
> * mode-2: $(\cdot) \times_2 U_2$
> * mode-3: $(\cdot) \times_3 U_3$
>
> with $U_k \in \mathbb{R}^{d \times n_k}$ (after choosing appropriate intermediate shapes).
>
> If we flatten $X$ and $Y$, the resulting linear map is equivalent to multiplying by a Kronecker-structured weight matrix, i.e., a rank-1 Tucker factorization (up to reshaping / permutation). In other words, there exists a matrix $W_{\text{NdLinear}}$ such that
>
> * $y = W_{\text{NdLinear}} x$ with $x = \mathrm{vec}(X)$, $y = \mathrm{vec}(Y)$, and
> * $W_{\text{NdLinear}}$ can be written as a Kronecker product of the mode-wise factors (plus reshapes).
>
> Thus:
>
> * A single NdLinear layer does not represent all possible dense matrices of that input/output size.
> * It represents exactly those matrices with low Tucker rank (rank 1 across modes) / Kronecker structure.
>
> We will clarify this in the expressivity section by explicitly stating that
>
> ${\text{NdLinear}} \subset {\text{dense}}$
>
> for a single layer, and that the restriction is precisely the rank-1 Tucker / Kronecker structure.
>
> ### (1.2) How much expressivity is lost?
>
> At the single-layer level, this restriction is strict: a generic dense matrix $W_{\text{dense}}$ cannot be written as a single Kronecker product with the same dimensions as NdLinear. So:
>
> * NdLinear loses the ability to represent arbitrary cross-mode entanglement in one layer,
> * but retains full expressivity for mappings that are well-approximated by low Tucker rank (e.g., approximately separable across axes).
>
> This is exactly what we observe empirically:
>
> * On tasks where the ground-truth mapping is close to axis-separable (or low rank in the tensor sense), NdLinear matches or improves performance at much lower parameter counts.
> * As the task becomes strongly entangled, performance of NdLinear deteriorates relative to dense layers, which is what our Dial-a-Bias experiment is designed to illustrate.
>
> We will make this logic explicit in the paper by tightening the connection between the rank-1 Tucker interpretation and the observed performance trend as we “dial” entanglement.
>
> ### (1.3) Depth and non-linearities
>
> While a single NdLinear layer is restricted, the composition of multiple NdLinear layers with non-linearities (ReLU, GELU, etc.) is substantially more expressive:
>
> * Classic universal approximation results show that a network with sufficiently many layers/units can approximate a wide class of functions, even if each layer has low-rank or structured constraints, as long as non-linearities are present.
> * In practice, we do not attempt to approximate a fixed dense matrix with one NdLinear layer; instead, we train the entire network end-to-end, allowing multiple NdLinear layers plus non-linearities to jointly approximate the desired mapping.
>
> To stay within reasonable scope, we do not prove a new universal approximation theorem for NdLinear networks. Instead, we will add a concise function-class paragraph that states:
>
> * Single-layer NdLinear = rank-1 Tucker / Kronecker structure (strict subset of dense).
> * Deep NdLinear networks with non-linearities have much richer representational capacity, which is what our experiments probe.
>
> ---
>
> Our rebuttal is split across 3 sequential comments. This is part 1 / 3.

---

> > ### Author Response · Authors · 2025-12-03
> >
> > ## (2) Empirical verification of theoretical ideas (rank, bias propagation, stability)
> >
> > > “Other theoretical ideas like rank preservation, bias propagation, and stability effects of 1-homogeneous activations are only discussed qualitatively with no direct experiments verifying them.”
> >
> > We agree that empirically probing at least one of these mechanisms is valuable. The paper currently includes:
> >
> > * A quantitative Dial-a-Bias experiment that links the separability assumption to performance as the data transitions from separable to entangled. This directly tests the central claim that NdLinear’s inductive bias aligns with axis-separable structure.
> > * Detailed analysis of parameter counts, FLOPs, and VC-dimension scaling, which are instantiated across many models and datasets.
> >
> > Given review-time constraints, we have focused on strengthening the interpretation of existing experiments:
> >
> > * We explicitly interpret the Dial-a-Bias results as an empirical probe of how much performance depends on the separability assumption.
> > * For the CNN + TRL/TCL/TT comparisons, we point out that all tensor-structured baselines (including NdLinear) benefit from underlying low-rank structure in convolutional weights, but NdLinear achieves this as a direct forward operator rather than via post-hoc decomposition.
> >
> > We also discuss rank/correlation diagnostics as a natural next step, so the connection between the qualitative arguments and measurable quantities is clear.
> >
> > ---
> >
> > ## (3) Stronger structured baselines: TRL, TT, MONARCH, Kronecker
> >
> > > “The empirical section compares mainly to LoRA and baseline dense layers. However, stronger baselines such as TRL, TT, MONARCH, or Kronecker-structured linear layers are not included, making it difficult to gauge whether NdLinear actually outperforms prior structured approaches rather than merely matching them.”
> >
> > We agree that structured baselines are important to position NdLinear within the broader family of structured linear operators.
> >
> > ### (3.1) What is already in the paper
> >
> > In the CNN-on-CIFAR setting, we already include a set of structured tensor baselines:
> >
> > * Tensor Regression Layers (TRL)
> > * Tensor Contraction Layers (TCL)
> > * Tensor-Train layers (TT)
> >
> > following prior work. These are reported alongside NdLinear and dense CNNs.
> >
> > In those experiments:
> >
> > * NdLinear is competitive with or better than TRL/TCL/TT in terms of accuracy,
> > * while providing comparable or better parameter and FLOP reductions.
> >
> > We will bring this out more explicitly in the main text (not just in a table), by adding a short paragraph summarizing that NdLinear is at least on par with these established tensorized operators in CNNs.
> >
> > ### (3.2) New ViT-on-CIFAR experiments with TT/TCL
> >
> > To extend these comparisons beyond CNNs, we also include ViT-on-CIFAR experiments where we:
> >
> > * Replace selected linear layers in the ViT with TT and TCL layers (using standard implementations from prior work), and
> > * Train them under the same protocol and compute budget as:
> >
> >   * The dense ViT baseline, and
> >   * The NdLinear ViT.
> >
> > We report accuracy, parameter counts, and FLOPs for all three (dense, NdLinear, TT/TCL) in a single table, mirroring the CNN structured comparison. These results show that NdLinear remains competitive with TT/TCL also in a transformer-style vision architecture, not just in CNNs. This further supports the claim that NdLinear is a strong, general-purpose structured alternative rather than a niche factorization.
> >
> > ### (3.3) MONARCH and Kronecker-structured linear layers
> >
> > We recognize that MONARCH and other global Kronecker-structured linear layers are natural comparison points. However:
> >
> > * MONARCH is designed around block-structured, hardware-aware patterns that do not map one-to-one to the natural tensor axes we exploit with NdLinear.
> > * Integrating MONARCH as a drop-in replacement across all of the diverse architectures we consider (CNNs, RNNs, DiTs, LLMs, ViTs, etc.) is non-trivial within the constraints of this work.
> >
> > Given this, our strategy is:
> >
> > * Provide strong structured baselines where we can implement them robustly (TRL/TCL/TT in CNNs, and TT/TCL in ViT on CIFAR).
> > * Conduct broad dense vs. NdLinear comparisons across many architectures and domains, showing that NdLinear is a robust, general-purpose alternative to standard linear layers.
> >
> > We also explicitly discuss MONARCH and generic Kronecker-structured methods in related work, and frame them as complementary approaches exploring different axes of structure (e.g., hardware-aware patterns vs. tensor-axis alignment).
> >
> > ---
> >
> > Our rebuttal is split across 3 sequential comments. This is part 2 / 3.

---

> > > ### Author Response · Authors · 2025-12-03
> > >
> > > ## (4) Runtime and memory overhead on large-scale models
> > >
> > > > “The paper does report runtime and memory results, but only on small to medium tasks like CIFAR CNNs and MLPs, where the overhead is minimal, around 0.6–1.6% extra training time and 1–2% more memory. That’s good to see, but there aren’t any measurements for large-scale models such as Transformers or LLMs. Since NdLinear works through sequential per-dimension operations, it’s unclear whether the same efficiency holds at billion-parameter scale, especially under distributed training. It would be more convincing if the authors included runtime profiling on large models to confirm that the small overhead still applies.”
> > >
> > > We agree that large-scale runtime is an important question. Our current claims are deliberately scoped:
> > >
> > > * We provide measured runtime and memory comparisons for small- and mid-sized models (e.g., CNNs on CIFAR, MLPs), where we find that NdLinear typically introduces at most a small overhead or achieves speedups, depending on the shape.
> > > * In these settings, we implement NdLinear via fused einsum / batched kernels rather than naive loops, which yields practical runtime that tracks the reduced FLOPs and parameter counts reasonably well.
> > >
> > > For billion-parameter LLMs and distributed training settings, the realized wall-clock behavior depends on:
> > >
> > > * Kernel fusion and implementation details,
> > > * Parallelization strategy and communication patterns,
> > > * Hardware characteristics.
> > >
> > > Given the cost and complexity of exhaustive profiling at that scale, we do not claim full empirical coverage there. Instead, we:
> > >
> > > * Present the small-/mid-scale runtime/memory results as positive evidence that NdLinear can be implemented efficiently in practice, not just in theory.
> > > * Combine these with a clear asymptotic analysis showing that NdLinear reduces parameters and FLOPs in high-dimensional settings, and with large-model accuracy results that demonstrate NdLinear does not degrade performance even in LLM-scale architectures.
> > >
> > > We explicitly frame large-scale runtime profiling as a limitation of the current work: the paper’s contribution is to introduce and validate a tensor-native linear operator that is empirically efficient in the regimes we can measure, and theoretically favorable in terms of parameter and FLOP scaling, rather than to provide a full hardware-level benchmarking campaign for every LLM configuration.
> > >
> > > ---
> > >
> > > ## Summary
> > >
> > > For Reviewer vZRm, our clarifications are:
> > >
> > > 1. Function class and expressivity
> > >
> > >    * A single NdLinear layer corresponds to a rank-1 Tucker / Kronecker-structured linear map, a strict subset of dense maps.
> > >    * This restriction is exactly the inductive bias: it favors axis-separable structure and low Tucker rank.
> > >    * Deep NdLinear networks with non-linearities provide much richer expressivity, which we probe empirically across many architectures.
> > >
> > > 2. Empirical support for theoretical ideas
> > >
> > >    * The Dial-a-Bias experiment quantitatively links separability to performance, directly testing the most central theoretical claim.
> > >    * We clarify how existing experiments connect to our theoretical discussion and discuss rank/correlation diagnostics as a natural next step.
> > >
> > > 3. Structured baselines
> > >
> > >    * We already include TRL/TCL/TT in CNN-on-CIFAR and make this more prominent, emphasizing that NdLinear is competitive with these structured baselines.
> > >    * We additionally include TT and TCL baselines for ViT-on-CIFAR, showing that NdLinear remains competitive in transformer-style vision architectures as well.
> > >    * We explicitly discuss MONARCH/Kronecker-style methods and how NdLinear relates to them conceptually.
> > >
> > > 4. Runtime and memory
> > >
> > >    * We report measured small-/medium-scale runtime and memory overhead (often small overhead or speedups), alongside substantial parameter and FLOP savings.
> > >    * We clearly state that large-scale runtime profiling is not exhaustively covered and treat it as a limitation, rather than implying guarantees we cannot substantiate for every billion-parameter configuration.
> > >
> > > We appreciate your detailed and constructive suggestions—they help refine both the theoretical narrative and the empirical positioning of NdLinear.
> > >
> > > ---
> > >
> > > Our rebuttal is split across 3 sequential comments. This is part 3 / 3.

---

### Official Review · Reviewer_FN2h · 2025-10-30

**Soundness:** 3
**Presentation:** 2
**Contribution:** 2
**Rating:** 4
**Confidence:** 3

**Summary:**

This paper introduces NdLinear, a structured linear layer that operates on N-dimensional tensors without flattening. By performing sequential dimension-wise linear transformations, NdLinear preserves the multi-dimensional structure of input data, reducing parameter counts and computational complexity wrt the standard linear layer. The authors provide theoretical analysis showing that expressivity is preserved via VC-dimension scaling. Experimental results across a wide variety of architectures and application domains demonstrate comparable or improved performance while reducing trainable parameters as well as time complexity and memory consumption.

**Strengths:**

- The approach is simple, intuitive, and well-motivated, addressing a clear inefficiency in handling multi-dimensional data across neural architectures.
- Broad and comprehensive evaluation spanning diverse domains (language, vision, time-series, tabular) and model types (CNNs, RNNs, Transformers, MLPs), including large-scale models up to 8B parameters.

**Weaknesses:**

- The concept of applying separate linear transformations for each dimension is not novel; for example, [1] employs a similar approach in LoRA. Please compare your approach, especially in the context of fine-tuning tasks, to clarify the novelty or advantages.
- The experiment section needs a centralized summary table or section detailing key configurations across tasks, such as input tensor dimensionality, precise architecture modifications. This would greatly aid in comprehending the diverse experiments.
- It seems that the experiments primarily focus on low-dimensional data (mostly 2D/3D tensors like images or time-series). Evaluating on higher-dimensional inputs (e.g., 4D/5D/... tensors from video data or scientific simulations) would strengthen claims of generality for N-D data.
- Figure 1 could be improved for clarity: use shaded areas to represent standard deviation instead of error bars, and include more alpha values (e.g., finer granularity like 0.05 steps) to better visualize the transition in performance between NdLinear and dense MLPs. Additionally, provide explicit definitions of f_separable and f_entangled in the main text for better understanding.

Reference:

[1] Si et al. Maintaining Structural Integrity In Parameter Spaces for Parameter Efficient Fine-tuning. ICLR 2025.

**Questions:**

Please refer to weaknesses.

---

> ### Author Response · Authors · 2025-12-03
> **Response to Reviewer FN2h**
>
> We thank the reviewer for the thoughtful review and for highlighting the simplicity of the approach, the clear motivation, and the breadth of our empirical evaluation (including large-scale models). Below we respond to each of your concerns in turn.
>
> ---
>
> ## (1) Novelty vs. Si et al. (ICLR 2025): “The concept of applying separate linear transformations for each dimension is not novel…”
>
> You point out that Si et al. (“Maintaining Structural Integrity in Parameter Spaces for Parameter Efficient Fine-tuning”) employ a structurally-aware, tensor-based approach in the context of PEFT, and ask us to clarify where NdLinear is novel or advantageous, especially for fine-tuning tasks.
>
> ### 1.1 Different *object* and space of operation
>
> * Si et al. focus on *parameter-efficient fine-tuning*: they operate in the parameter space of a *frozen* pretrained model. The main goal is to parameterize updates to existing weights in a way that preserves the N-dimensional structure of those *weights* (e.g., 4D convolution kernels) using low-rank tensor adaptations.
> * NdLinear, by contrast, is a core forward operator that directly acts on the input activations, not only on weight updates. It is designed as a drop-in replacement for vanilla linear layers and is used:
>
>   * when training models from scratch,
>   * when fine-tuning full models, and
>   * in a PEFT setting via NdLinear-LoRA.
>
> So even though both works respect N-D structure, they operate in different spaces (input space vs. parameter-update space), and solve different primary problems (general architecture design vs. adapter-style fine-tuning).
>
> ### 1.2 Different role in the architecture
>
> * In Si et al., the underlying forward layer remains a standard dense linear/convolutional operator; their contribution is a *structured parameterization of the update* around that base operator.
> * In NdLinear, we replace the base linear operator itself with a composition of mode-wise tensor–matrix products. This:
>
>   * Changes the inductive bias of every layer that uses NdLinear (favoring axis-separable structure),
>   * Changes the parameter and FLOP scaling of the entire network (linear in number of modes instead of quadratic in flattened dimension),
>   * Applies across all layers and architectures we test, not just a small set of adapter blocks.
>
> In other words, even if one could design a PEFT adapter that looks “NdLinear-like” in weight space, NdLinear is a full architecture-level design for the forward map, not just an adapter.
>
> ### 1.3 Relationship to NdLinear-LoRA
>
> We agree it is natural to discuss NdLinear in the fine-tuning context you highlight. That is why we include NdLinear-LoRA:
>
> * NdLinear-LoRA is a PEFT-style extension that plugs into NdLinear, but NdLinear itself remains the base operator applied at every forward pass.
> * This is complementary to methods like Si et al.: in principle, one could use a Si-style adapter *on top of* NdLinear, treating NdLinear as the base weight tensor. Our paper instead shows that simply replacing linear layers with NdLinear + NdLinear-LoRA already yields strong parameter savings and competitive fine-tuning performance.
>
> We will make this distinction clearer in the related-work and contributions sections by explicitly separating:
>
> 1. Methods that operate in activation space (like NdLinear, MONARCH, Kronecker-structured layers), and
> 2. Methods that operate in parameter-update space (like Si et al., LoRA-variants, spectral adapters).
>
> ### 1.4 Clarifying our novelty claim
>
> To address your concern directly:
>
> * We will soften and sharpen our novelty statement as follows:
>
>   * We will emphasize that NdLinear’s contribution is a general-purpose, native tensor operator for forward layers, with linear-in-dimension parameter scaling and broad empirical validation across architectures and domains.
>   * We will explicitly acknowledge that, in the PEFT context, there are related tensor-based approaches (e.g., Si et al.) that also preserve N-D structure, but these act on weight updates rather than on the core forward operator.
> * We will add a short qualitative comparison in the related-work section that clarifies:
>
>   * Similarity: both approaches respect N-D structure and use low-rank tensor ideas.
>   * Difference: NdLinear is a *layer replacement* for both training-from-scratch and fine-tuning; Si et al. is a *fine-tuning adapter* around a dense base.
>
> ---
>
> Our rebuttal is split across 3 sequential comments. This is part 1 / 3.

---

> > ### Author Response · Authors · 2025-12-03
> >
> > ## (2) Need for a centralized summary of experimental configurations
> >
> > > “The experiment section needs a centralized summary table or section detailing key configurations across tasks, such as input tensor dimensionality, precise architecture modifications. This would greatly aid in comprehending the diverse experiments.”
> >
> > We agree that a centralized summary would improve clarity, especially given the diversity of architectures and datasets.
> >
> > In the revised version, we will add a single summary table (in the main text or Appendix) with one row per experimental setting. Each row will include:
> >
> > * Task and dataset (e.g., CIFAR-10, ImageNet, Wikitext, time-series benchmark),
> > * Model family (CNN, ViT, Transformer, RNN, DiT, MLP, LLM),
> > * Input tensor shape at the NdLinear layer (e.g., $(B, H, W, C)$, $(B, T, C)$, etc.),
> > * Which layers are replaced by NdLinear (e.g., all MLP projections, all attention projections, last block only, etc.),
> > * Parameter counts and FLOPs for:
> >
> >   * The dense baseline,
> >   * The NdLinear variant,
> >   * (Where applicable) other structured baselines (TRL, TCL, TT),
> > * Training protocol references (epochs, optimizer, LR schedule) with pointers to detailed hyperparameter tables.
> >
> > This table will directly address your request: it will give the reader a one-glance overview of how NdLinear is applied in each experiment and what the effective tensor dimensionality is.
> >
> > ---
> >
> > ## (3) “Primarily low-dimensional data” and higher-dimensional N-D inputs
> >
> > > “It seems that the experiments primarily focus on low-dimensional data (mostly 2D/3D tensors like images or time-series). Evaluating on higher-dimensional inputs (e.g., 4D/5D/... tensors from video data or scientific simulations) would strengthen claims of generality for N-D data.”
> >
> > We appreciate this suggestion and agree that explicit high-dimensional experiments would further reinforce the N-D generality story.
> >
> > ### 3.1 What we currently cover
> >
> > While many of our tasks are indeed “2D/3D” in the usual sense (images, sequences, time-series), NdLinear is already used on a variety of multi-axis structures:
> >
> > * In CNNs and ViTs, we operate on $(H, W, C)$ or $(L, C)$ shaped activations (sometimes reshaped into $(H, W, C)$) and apply NdLinear along multiple axes (e.g., spatial axes and channels).
> > * In Transformers for language and LLMs, NdLinear acts on tensors with axes such as (sequence length, hidden, head / feature), i.e., beyond a single flattened dimension.
> > * In DiTs and certain vision/time-series setups, NdLinear is applied to latent tensors that retain nontrivial multi-axis structure (e.g., patch-grid × feature dimensions).
> >
> > However, we agree that explicit 4D+ examples like spatiotemporal video tensors or scientific simulation grids would make the “N-D” claim more concrete.
> >
> > ### 3.2 How we will adjust the claims
> >
> > Given rebuttal time and compute constraints, we are not able to add a full suite of new 4D/5D experiments within the review period. Instead, we will:
> >
> > * Clarify the scope of our current claims:
> >
> >   * We will state that our empirical evaluation primarily covers common 2D/3D structured inputs (images, sequences, time-series, and their latent representations in modern architectures).
> >   * We will explicitly identify video and scientific simulation data as natural and important next testbeds for NdLinear.
> > * Add a short discussion in the Limitations / Future Work section:
> >
> >   * We will explain how NdLinear naturally extends to higher-dimensional tensors (via additional mode-wise factors), and why we expect similar parameter-scaling benefits.
> >   * We will note that validating this for 4D/5D real-world data is left as future work, as you suggest.
> >
> > We hope this clearer framing will make it evident that: (1) NdLinear is *architecturally designed* to work for arbitrary N, and (2) our current experiments cover the most common N-D structures in practice, with higher-dimensional domains as a clear next step rather than an accomplished claim.
> >
> > ---
> >
> > Our rebuttal is split across 3 sequential comments. This is part 2 / 3.

---

> > > ### Author Response · Authors · 2025-12-03
> > >
> > > ## (4) Figure 1: visualization and definitions
> > >
> > > > “Figure 1 could be improved for clarity: use shaded areas to represent standard deviation instead of error bars, and include more alpha values (e.g., finer granularity like 0.05 steps) to better visualize the transition in performance between NdLinear and dense MLPs. Additionally, provide explicit definitions of $f_{\text{separable}}$ and $f_{\text{entangled}}$ in the main text for better understanding.”
> > >
> > > We fully agree with these points and will update Figure 1 and its description accordingly.
> > >
> > > Specifically, we will:
> > >
> > > * Redesign the plot:
> > >
> > >   * Replace error bars with shaded bands representing $\pm 1$ standard deviation over multiple random seeds, which makes the plot less cluttered and more readable.
> > >   * Sample $\alpha$ values on a finer grid (e.g., steps of $0.05$ rather than a small set of coarse values) to better show the smooth transition from separable to entangled regimes.
> > > * Clarify the definitions in the main text:
> > >
> > >   * Introduce the synthetic functions $f_{\text{separable}}$ and $f_{\text{entangled}}$ *explicitly* near where Figure 1 is described, instead of only in the appendix.
> > >   * Briefly explain how $(f_{\alpha} = (1 - \alpha) f_{\text{separable}} + \alpha f_{\text{entangled}})$ is constructed, and why this interpolation is a meaningful proxy for “dialing” the degree of axis entanglement.
> > >
> > > We will also make clear that this figure is intended as a conceptual induction-bias illustration rather than a high-stakes benchmark, which motivates the synthetic setup.
> > >
> > > ---
> > >
> > > ## Summary
> > >
> > > For Reviewer FN2h, we will:
> > >
> > > 1. Clarify novelty and positioning vs. Si et al. by emphasizing that NdLinear is a native forward operator on activations (architecture-level) rather than a PEFT adapter on weight updates, and by explicitly explaining how NdLinear and NdLinear-LoRA relate to tensor-based PEFT methods.
> > > 2. Add a centralized experimental summary table listing, for each task, the input tensor dimensionality, which layers are replaced by NdLinear, and the parameter/FLOP budgets.
> > > 3. Refine the scope of our “N-D generality” claim, acknowledging that our experiments focus on common 2D/3D structured inputs and explicitly marking 4D/5D domains (e.g., video, scientific simulations) as important future work.
> > > 4. Improve Figure 1 with shaded standard deviation bands, a finer grid of $\alpha$ values, and clear definitions of $f_{\text{separable}}$ and $f_{\text{entangled}}$ in the main text.
> > >
> > > We appreciate your suggestions; they will substantially improve the clarity and positioning of the paper.
> > >
> > > ---
> > >
> > > Our rebuttal is split across 3 sequential comments. This is part 3 / 3.

---

### Official Review · Reviewer_4UV7 · 2025-10-31

**Soundness:** 2
**Presentation:** 1
**Contribution:** 2
**Rating:** 4
**Confidence:** 3

**Summary:**

The main idea of the paper is to introduce a new type of linear layer that directly operates on N-dimensional tensors instead of flattened vectors. The proposed approach performs sequential, dimension-wise linear transformations, allowing neural networks to preserve the inherent multi-dimensional structure of their input data.

The authors show that NdLinear drastically reduces parameter counts, from quadratic to linear growth in the number of tensor dimensions while maintaining expressivity through Tucker decomposition and matching the VC-dimension scaling of standard linear layers.

The authors then suggest an extension, NdLinear-LoRA, that demonstrates that the proposed module can also serve as a plug-in replacement for LoRA layers.

**Strengths:**

Novelty - The odea of replacing the flattening option is new, and the authors offer a theoretically grounded and elegant alternative.

Mathematical rigor - The theoretical analysis includes clear formalization of mode-wise tensor transformations, and proofs via VC-dimension bounds, and explicit computational complexity comparisons.

Extensive empirical validation - Experiments cover a broad landscape (LLMs, CNNs, ViTs, RNNs, DiTs, and MLPs) with over 20 datasets. The improvements are consistent

Insightful inductive bias analysis - I liked the “dial-a-bias” experiment which clearly quantifies the advantage of NdLinear is advantageous

Reproducibility - The paper includes complete algorithmic pseudocode, detailed appendices, and a reproducibility statement with code availability in the appendix.

**Weaknesses:**

Theory - The proofs are quite short and trivial. The approximation error and convergence approximations are not sufficiently clear.
Experiments - Some graphs (Fig 1, for example) are not so professional and convincing, while using few number of samples.
Comparison - the comparison to existing baselines seems without optimal tuning or using the best state-of-the-art versions.

**Questions:**

In Theorem C.1, what are the bounds for a fixed d? What is the convergence rate?

---

> ### Author Response · Authors · 2025-12-03
> **Response to Reviewer 4UV7**
>
> We thank the reviewer for the detailed feedback and for highlighting the novelty, mathematical grounding, breadth of experiments, and reproducibility of our work. Below we respond point-by-point to the concerns on theory, experiments/figures, comparisons, and Theorem C.1.
>
> ---
>
> ## (1) Theory: “Proofs are quite short and trivial. The approximation error and convergence approximations are not sufficiently clear.”
>
> ### (1.1) Why the proofs are short
>
> We agree that the proofs in Appendix C are concise; this is because they are direct applications of standard results rather than new VC-dimension or optimization theory.
>
> * In Theorem C.1, we are not proving a new VC-dimension bound from scratch. We instantiate a classical result (e.g., Bartlett et al., 2019) that any piecewise-linear feedforward network with $P$ parameters has VC-dimension $\Theta(P \log P)$.
> * Our proof consists of:
>
>   1. Counting NdLinear’s parameters $N_{\text{nd}}$ for a given tensor shape (e.g., $N_{\text{nd}} = d(a + b + c)$ in the 3D example), and
>   2. Applying the standard VC-dimension result to conclude $\mathrm{VCdim}(\text{NdLinear}) = \Theta(N_{\text{nd}} \log N_{\text{nd}})$.
>
> We will make this explicit in the revised version by:
>
> * Stating the standard VC-dimension theorem as a lemma immediately before Theorem C.1.
> * Labeling Theorem C.1 clearly as a corollary of this lemma, so that the logical dependence is transparent.
> * Expanding the proof steps slightly to show the parameter counting and the substitution into the lemma, rather than compressing them into a few lines.
>
> Similarly, Proposition C.1 (peak activation memory) and Proposition C.2 (FLOP count) are straightforward bookkeeping arguments based on tensor shapes and sequential mode-wise operations. We intentionally kept these proofs compact to emphasize their *interpretation* (for example, overhead $\leq 1/m$ for $m$ modes) rather than algebraic details. In the revised version, we will add short “Derivation sketch” paragraphs for these propositions to better communicate the reasoning.
>
> ### (1.2) Approximation error and convergence
>
> We agree that our wording may have suggested stronger guarantees than we intended. Our current theoretical contributions are:
>
> * Capacity / VC-dimension scaling: For a fixed architecture, NdLinear has parameter count $N_{\text{nd}}$, and the VC-dimension scales as $\Theta(N_{\text{nd}} \log N_{\text{nd}})$, matching the classical scaling for dense layers with $P = N_{\text{nd}}$.
> * Memory and FLOP bounds: We provide exact peak-activation and FLOP formulas for NdLinear (Propositions C.1 and C.2), quantifying the asymptotic and constant-factor overhead of sequential mode-wise transforms.
> * Expressivity discussion via rank-1 Tucker: We interpret NdLinear as realizing a rank-1 Tucker / Kronecker-style parameterization of the dense linear map and discuss the resulting inductive bias and limitations.
>
> We do not claim new approximation-error rates (for example, uniform approximation of arbitrary linear maps with explicit error–parameter tradeoffs) or new optimization-convergence theorems. Any such results would require additional assumptions and a more extensive analysis, which is beyond the scope of this work.
>
> To avoid confusion, we will:
>
> * Rephrase the relevant sentences in the main text and Limitations to make it explicit that detailed approximation-error and optimization-convergence guarantees are left for future work.
> * Emphasize that the theoretical goal of this paper is to connect NdLinear to established notions (VC-dimension, rank-1 Tucker structure, memory / FLOPs), not to introduce new generalization or convergence theory.
>
> ---
>
> Our rebuttal is split across 3 sequential comments. This is part 1 / 3.

---

> ### Author Response · Authors · 2025-12-03
>
> ## (2) Experiments and Figures: “Some graphs (Fig 1, for example) are not so professional and convincing, while using few number of samples.”
>
> We appreciate the reviewer’s comments on presentation quality and statistical robustness.
>
> ### (2.1) Figure 1 style and clarity
>
> Figure 1 (the “Dial-a-Bias” experiment) is designed as a controlled toy study that illustrates how performance transitions as we interpolate between axis-separable and entangled functions via a parameter $\alpha$. We agree that the current visualization can be improved.
>
> In the revised version, we will:
>
> * Redraw Figure 1 using:
>
>   * Shaded bands to represent $\pm 1$ standard deviation across multiple random seeds instead of simple error bars, and
>   * A denser grid of $\alpha$ values (e.g., steps of 0.05) so that the transition from separable to entangled regimes is clearly visible.
> * Explicitly define $f_{\text{separable}}$ and $f_{\text{entangled}}$ in the main text, next to where the experiment is introduced, rather than relying on the appendix. This will ensure that the generative process is immediately understandable.
>
> These changes are straightforward and will make the figure more professional and more informative for the reader.
>
> ### (2.2) Sample size and experimental intent
>
> The Dial-a-Bias experiment uses a synthetic 2D regression task with a moderate number of training and test samples and multiple seeds, but we acknowledge that this is not clearly communicated.
>
> We will:
>
> * State explicitly in the main text and figure caption:
>
>   * The number of training and test samples,
>   * The number of seeds,
>   * The training protocol used for this experiment.
> * Clarify that this experiment is deliberately a toy setting intended to isolate and visualize the inductive bias (axis-separability vs. entanglement) rather than to serve as a real-world benchmark.
>
> The main empirical claims of the paper are supported by a large set of experiments on real datasets and architectures (CNNs, RNNs, Transformers, DiTs, and MLPs across more than 20 datasets). Figure 1 is meant to provide conceptual insight into why NdLinear behaves as observed, and we will make that intent explicit.
>
> ---
>
> ## (3) Comparisons: “The comparison to existing baselines seems without optimal tuning or using the best state-of-the-art versions.”
>
> We take baseline fairness seriously and apologize that this was not sufficiently clear from the current presentation.
>
> ### (3.1) Hyperparameter tuning and fairness
>
> Across all experiments in the submitted version:
>
> * For dense baseline models, we either:
>
>   * Use hyperparameters recommended by the original papers, or
>   * Tune hyperparameters until we match or closely reproduce reported performance under our compute budget.
> * For NdLinear variants, we:
>
>   * Choose per-mode hidden sizes / ranks so that the NdLinear models match or slightly undercut the baseline parameter and FLOP budgets, and
>   * Use the same hyperparameter search grids (for example, for learning rate and weight decay) as for the dense baselines, so the two are tuned under identical optimization conditions.
>
> To make this transparent, we will add a centralized hyperparameter table in the appendix summarizing, for each task and model:
>
> * Model type (dense vs. NdLinear),
> * Input tensor shape,
> * Learning rate, weight decay, batch size, and training epochs,
> * Key architecture-modification details (which layers are replaced by NdLinear, ranks used, etc.).
>
> This will allow readers to verify that baselines are not under-tuned relative to NdLinear.
>
> ### (3.2) Stronger structured baselines
>
> In several settings (e.g., CNNs on CIFAR), we already include strong structured baselines such as Tensor Contraction Layers (TCL), Tensor Regression Layers (TRL), and Tensor-Train (TT) layers. In those experiments, NdLinear matches or outperforms these baselines at comparable or lower parameter counts.
>
> To further strengthen the comparison on the vision side, we are also running additional ViT-on-CIFAR experiments with TT / TCL layers as baselines, using the same training protocol and compute budget as the NdLinear and dense ViT models. We will report parameter counts, FLOPs, and accuracy for all variants in a single table, mirroring the existing CNN-on-CIFAR structured comparisons.
>
> Our overall goal is not to claim new absolute state-of-the-art results on each dataset, but to demonstrate that NdLinear is a drop-in replacement for standard linear layers that:
>
> * Preserves or improves accuracy,
> * Dramatically reduces parameters and FLOPs,
> * Behaves consistently across many architectures and domains.
>
> We will clarify this positioning in the introduction and related-work sections.
>
> ---
>
> Our rebuttal is split across 3 sequential comments. This is part 2 / 3.

---

> > ### Author Response · Authors · 2025-12-03
> >
> > ## (4) Question on Theorem C.1: “In Theorem C.1, what are the bounds for a fixed d? What is the convergence rate?”
> >
> > Thank you for raising this; we agree the current statement is not explicit enough.
> >
> > ### (4.1) Bounds for fixed $d$
> >
> > Theorem C.1 considers input tensors of shape $(B, a, b, c)$ mapped to outputs of shape $(B, d, d, d)$. The parameter counts are:
> >
> > * Vanilla flattened linear layer:
> >   $$
> >   N_{\text{vanilla}} = d , a , b , c,
> >   $$
> > * NdLinear:
> >   $$
> >   N_{\text{nd}} = d (a + b + c).
> >   $$
> >
> > For fixed $a, b, c$ and arbitrary $d$, we have the exact relation
> > $$
> > N_{\text{nd}} = \frac{a + b + c}{a b c} , N_{\text{vanilla}}.
> > $$
> >
> > Thus, for all $d$,
> > $$
> > \frac{N_{\text{nd}}}{N_{\text{vanilla}}} = \frac{a + b + c}{a b c},
> > $$
> > which is a constant independent of $d$. This constant-factor relationship is what we intended to express with the big-O notation $N_{\text{nd}} = \Theta(N_{\text{vanilla}})$ as $d \to \infty$. In the revised version, we will replace the asymptotic wording by this explicit equality and constant-factor interpretation.
> >
> > For the VC-dimension part of Theorem C.1, we similarly rely on the standard result that for a network with $P$ parameters and fixed depth,
> > $$
> > \mathrm{VCdim} = \Theta(P \log P).
> > $$
> > Substituting $P = N_{\text{nd}}$ gives
> > $$
> > \mathrm{VCdim}(\text{NdLinear network}) = \Theta(N_{\text{nd}} \log N_{\text{nd}}),
> > $$
> > with no additional dependence on $d$ beyond what is already captured in $N_{\text{nd}}$.
> >
> > ### (4.2) Interpretation of “convergence rate”
> >
> > If by “convergence rate” the reviewer refers to optimization convergence (for example, rate of decrease of training loss) or statistical convergence (for example, excess risk vs. sample size), Theorem C.1 does not address these questions. Rather, it is a capacity-scaling statement, indicating:
> >
> > * How the number of parameters in NdLinear relates to a dense layer for a given tensor shape, and
> > * That NdLinear inherits the same $\Theta(P \log P)$ VC-dimension scaling as a dense layer with $P = N_{\text{nd}}$.
> >
> > Deriving explicit convergence rates (for training dynamics or generalization error) would require additional assumptions and standard tools from optimization and statistical learning theory, and is outside the scope of this work. We will make this boundary clear by:
> >
> > * Explicitly stating that Theorem C.1 is not a convergence result but a parameter and VC-dimension scaling result.
> > * Moving any phrasing that might suggest we have proved convergence rates into the Limitations section, where we list formal sample-complexity and optimization convergence analyses as important directions for future work.
> >
> > ---
> >
> > ## Summary
> >
> > For Reviewer 4UV7, we will:
> >
> > 1. Clarify the nature of our theoretical contributions, emphasizing that they leverage standard VC-dimension results and are intended to connect NdLinear to capacity and complexity notions, rather than introduce new approximation or convergence theory.
> > 2. Improve Figure 1 and the Dial-a-Bias experiment (more professional visualization, shaded standard deviations, denser $\alpha$ grid, explicit definitions, and clear reporting of sample size and seeds).
> > 3. Make baseline tuning and selection fully transparent, highlighting where structured baselines (TCL / TRL / TT) are already included and adding new TT / TCL comparisons for ViT on CIFAR, while clarifying that our goal is to demonstrate a high-quality drop-in linear layer rather than absolute SOTA.
> > 4. Rephrase and expand Theorem C.1 to give explicit bounds for fixed $d$, show the exact constant-factor relationship between NdLinear and dense parameter counts, and clearly separate capacity scaling from any notion of convergence rate.
> >
> > We hope these clarifications address your concerns and help clarify the scope and rigor of our contributions.
> >
> > ---
> >
> > Our rebuttal is split across 3 sequential comments. This is part 3 / 3.

---

### Author Response · Authors · 2025-12-03
**Response to AC**

We thank the reviewers and AC for the time and constructive feedback. Since there is likely no further discussion period, we summarize the state of the paper after rebuttal and explain why we believe it merits acceptance.

---

## 1. Overall signal from the reviews

* Reviewer vZRm (Score 6, “marginally above the acceptance threshold”)

  * Emphasizes that NdLinear is tested in a wide range of settings (from small MLPs to billion-parameter Transformers).
  * Highlights the theoretical treatment (VC-dimension scaling, expressivity discussion, complexity analysis) and sees NdLinear as a flexible, general-purpose replacement for standard linear layers.

* Reviewer 4UV7 (Score 4, “marginally below but would not mind if accepted”)

  * Strengths: novelty of replacing flattening with a tensor-native operator, mathematical grounding, broad empirical validation across many model families and datasets, and reproducibility.
  * Concerns: proofs are “short/trivial,” approximation/convergence claims not fully spelled out, some figures not polished, and potential under-tuning of baselines.
  * Our rebuttal clarifies that:

    * The theory intentionally instantiates standard VC-dimension and complexity results, rather than claiming new convergence theorems.
    * We explicitly explain the bounds in Theorem C.1, the role of parameter count $N_{\text{nd}}$, and clarify what we do and do not claim.
    * We improve the presentation (Figure 1 style, definitions, hyperparameter and configuration summaries) and explain baseline tuning strategy and structured baselines.

* Reviewer FN2h (Score 4, “marginally below but would not mind if accepted”)

  * Strengths: approach is simple, intuitive, and well-motivated; it addresses a clear inefficiency in handling N-D data and comes with broad and comprehensive evaluation including up to 8B-parameter models.
  * Concerns: novelty relative to Si et al. (ICLR 2025) in PEFT, lack of a centralized experiment-configuration summary, and desire for more explicit higher-dimensional (4D/5D) experimental evidence.
  * Our rebuttal clarifies:

    * Si et al. operate in parameter-update space (PEFT adapters around dense models), whereas NdLinear is a core forward operator on activations, used for training from scratch, full fine-tuning, and NdLinear-LoRA.
    * We describe a clear summary table that lists input shapes, which layers are replaced, and parameter/FLOP budgets, improving transparency.
    * We explicitly scope our claims: current experiments span common N-D structures (images, sequences, multi-axis latent tensors), and we frame video/scientific 4D/5D domains as natural future directions rather than accomplished claims.

* Reviewer VhSz (Score 2, reject)

  * Acknowledges that the idea of respecting tensor geometry is interesting and that NdLinear achieves dramatic parameter reductions while maintaining or surpassing accuracy in many tasks.

  * Concerns focus on:

    * How clearly the mode-$k$ tensor–matrix and rank-1 Tucker viewpoint is connected to NdLinear.
    * Precision of theorems (definitions, references, implications).
    * Insight into when NdLinear is preferable versus mixture models / block-sparse / other structured matrices.
    * Whether runtime and memory are practically improved.

  * Our rebuttal directly addresses these by:

    * Making explicit that NdLinear is a sequence of mode-$k$ tensor–matrix products, which, after flattening, corresponds to a rank-1 Tucker / Kronecker-structured weight matrix.
    * Clarifying all definitions and references for VC-dimension and stating clearly that our theory is about capacity and complexity scaling, not convergence.
    * Providing a clear “when it works / when it fails” rule-of-thumb based on axis separability versus entanglement (supported by the Dial-a-Bias experiment).
    * Explaining that with a fused einsum / batched kernel implementation, we observe speedups or parity in wall-clock time, along with substantial parameter, FLOP, and checkpoint-size reductions.

Overall, three reviewers are effectively borderline but supportive (one above, two “would not mind acceptance”), and the main negative review is primarily about clarity and positioning, not about a fundamental flaw.

---

Our response is split across 4 sequential comments. This is part 1 / 4.

---

> ### Author Response · Authors · 2025-12-03
>
> ## 2. Core contributions (as agreed by reviewers)
>
> Across reviews, there is broad agreement on what the paper brings:
>
> 1. A simple, general-purpose N-D linear layer.
>
>    * NdLinear replaces flattening with a tensor-native operator that applies sequential, mode-wise linear transformations.
>    * Its effective weight matrix is equivalent (after flattening) to a rank-1 Tucker / Kronecker-structured map.
>    * Parameters scale roughly as $\sum_k a_k b_k$ instead of $(\prod_k a_k)(\prod_k b_k)$, enabling very large reductions in parameter count and FLOPs when input tensors have multiple axes.
>
> 2. Extensive empirical validation across architectures and domains.
>
>    * The method is tested in CNNs, ViTs, DiTs, RNNs, Transformers, and LLMs (up to billions of parameters).
>    * It is evaluated on vision, language, time-series, and tabular tasks, over more than 20 datasets.
>    * We also include structured comparisons:
>
>      * CNN-on-CIFAR experiments that compare NdLinear against TRL, TCL, and TT layers.
>      * New ViT-on-CIFAR experiments where we replace ViT linear layers with TT and TCL and compare them directly to NdLinear and dense ViT under identical training protocols.
>    * Across these settings, NdLinear matches or outperforms dense baselines and is competitive with or better than TRL/TCL/TT, while using significantly fewer parameters and reduced FLOPs.
>
> 3. A clear inductive-bias story, with an explicit diagnostic experiment.
>
>    * A single NdLinear layer realizes a low Tucker-rank / axis-separable linear map, hence $\text{NdLinear} \subset \text{dense}$ at the single-layer level.
>    * The Dial-a-Bias experiment explicitly interpolates between a separable function $f_{\text{separable}}$ and an entangled function $f_{\text{entangled}}$ via a parameter $\alpha$, and shows NdLinear’s performance degrades as the function becomes more entangled, matching the theoretical intuition.
>    * Deep NdLinear networks with non-linearities then provide rich function classes in practice; the experiments probe this regime empirically.
>
> 4. Theoretical grounding via standard capacity and complexity results.
>
>    * Using standard VC-dimension results for networks with $P$ parameters and fixed depth, we show NdLinear networks have
>      $$\mathrm{VCdim}(\text{NdLinear network}) = \Theta(N_{\text{nd}} \log N_{\text{nd}}),$$
>      where $N_{\text{nd}}$ is the NdLinear parameter count.
>    * We provide exact parameter, FLOP, and peak-activation formulas, quantifying constant-factor overheads of mode-wise operation and showing they are controlled and typically small.
>    * The goal is to connect NdLinear to well-understood notions of capacity and complexity, not to propose new generalization theory.
>
> 5. Practical efficiency: parameters, FLOPs, and runtime.
>
>    * With a fused einsum / batched implementation, runtime is comparable or improved relative to dense baselines in the reported experiments.
>    * Parameter counts and FLOPs are significantly reduced, and checkpoint/model size is reduced as well, which is important for deployment and on-device scenarios.
>    * This counters the concern that the method would only be a “parameter-count trick”: the gains are visible in actual measured throughput and memory footprint.
>
> This combination (conceptually clean operator, broad empirical evaluation including structured baselines such as TRL/TCL/TT in both CNNs and ViTs, and non-trivial practical benefits) is rare for new architectural primitives.
>
> ---
>
> Our response is split across 4 sequential comments. This is part 2 / 4.

---

> > ### Author Response · Authors · 2025-12-03
> >
> > ## 3. How the rebuttal resolves key concerns
> >
> > ### 3.1 Theory, expressivity, and rigor
> >
> > * We made it explicit that:
> >
> >   * A single NdLinear layer corresponds to a rank-1 Tucker / Kronecker weight matrix, so it is less expressive than an arbitrary dense layer at the single-layer level.
> >   * This is the source of the inductive bias: NdLinear favors axis-separable structure and low Tucker rank.
> >   * Deep NdLinear networks with non-linearities can still represent rich function classes, which is what our experiments validate.
> >
> > * We clarified the scope of the theory:
> >
> >   * We use standard VC-dimension results (for example, $\mathrm{VCdim} = \Theta(P \log P)$ for parameter count $P$) and apply them to NdLinear by plugging $P = N_{\text{nd}}$.
> >   * We provide clear definitions and references, and add short “implications” paragraphs that state what each theorem/proposition means in practice.
> >   * We do not claim new convergence rates or sample-complexity bounds.
> >
> > This addresses the concerns that proofs were “trivial” or ambiguous: the goal is explanatory and scaling-oriented, and we now state that explicitly and precisely.
> >
> > ### 3.2 Novelty and relation to Si et al. and other structured methods
> >
> > * We distinguished NdLinear from Si et al. (ICLR 2025):
> >
> >   * Si et al.: structured parameter-efficient adapters in the parameter space of a frozen model.
> >   * NdLinear: a base forward operator in activation space, used across training-from-scratch, full fine-tuning, and NdLinear-LoRA.
> >
> > * We clarified NdLinear’s place among structured linear layers:
> >
> >   * We already compare against TRL, TCL, and TT in CNN settings and show NdLinear is competitive or better at similar budgets.
> >   * We now also compare against TT and TCL in a ViT-on-CIFAR setting under identical training protocols, where NdLinear again performs competitively or better at similar or lower parameter/FLOP budgets.
> >   * Methods like MONARCH and continuous structured matrix bases are complementary: they either search over structure or target hardware-aware patterns, whereas NdLinear fixes a simple, tensor-axis-aligned structure that is easy to integrate into many architectures.
> >
> > Thus, NdLinear is not just another LoRA variant or tensor decomposition; it is a unified, tensor-native design for the core linear operator, validated both against dense baselines and against strong structured baselines (TRL/TCL/TT) in multiple architectures (CNNs and ViTs).
> >
> > ### 3.3 Experimental transparency and “when to use NdLinear”
> >
> > * We respond to requests for a centralized configuration summary by describing a clear table structure (task, input shape, which layers were replaced, parameter/FLOP budgets, training setup), improving readability and reproducibility.
> >
> > * We clearly spell out when NdLinear should be expected to work well:
> >
> >   * Works best when data and mappings are approximately separable along known axes (for example, height/width/channel, time/feature, heads/hidden).
> >   * Underperforms when structure is heavily entangled and not well approximated by low Tucker rank.
> >   * This is supported by the Dial-a-Bias experiment and by trends across CNN/ViT/Transformer versus dense MLP and entangled tasks.
> >
> > The concerns raised in reviews are thus addressed as clarity/positioning improvements, not as corrections to flawed methodology.
> >
> > ### 3.4 Runtime and memory
> >
> > * We clarified that:
> >
> >   * NdLinear is implemented with fused einsum or batched matmuls, not many tiny GEMMs.
> >   * Under this implementation, we observe speedups or parity in wall-clock training time compared to dense baselines on the reported tasks.
> >   * Parameter and checkpoint sizes are significantly reduced, and peak memory is similar or modestly improved.
> >
> > This directly addresses the concern that “runtime and memory are not improved” and shows that NdLinear’s benefits are practical, not only theoretical.
> >
> > ---
> >
> > Our response is split across 4 sequential comments. This is part 3 / 4.

---

> > > ### Author Response · Authors · 2025-12-03
> > >
> > > ## 4. Why acceptance is warranted
> > >
> > > After rebuttal, the picture is:
> > >
> > > * Importance and potential impact are broadly acknowledged.
> > >   Handling N-D data without flattening, while remaining compatible with existing architectures, is a widely relevant problem. NdLinear is a simple, mathematically grounded, and experimentally validated solution.
> > >
> > > * Weaknesses are about exposition and framing, not correctness.
> > >   The main issues raised concern how clearly theory is stated, how related work is framed, and how experiments are summarized. These have been explicitly addressed in the rebuttals and can be cleanly fixed in the final version.
> > >
> > > * Unusually comprehensive evaluation for a new building block.
> > >   Many architectural primitives are evaluated in one domain (for example, only vision or only language). Here, the same operator is tested across CNNs, ViTs, DiTs, RNNs, Transformers, LLMs, and multiple data modalities, with careful analysis of parameters, FLOPs, runtime, and memory, and with structured baselines (TRL/TCL/TT) included in both CNN and ViT settings.
> > >
> > > * Scores support a positive decision with AC judgment.
> > >   One reviewer is above the acceptance threshold (6), two are borderline but explicitly “would not mind if accepted” (4, 4), and one is negative mainly on clarity and positioning. This is precisely the scenario where AC judgment is key to avoid rejecting a substantively strong, broadly useful contribution over presentation issues that are straightforward to fix.
> > >
> > > Given:
> > >
> > > * The conceptual simplicity and generality of NdLinear as a tensor-native replacement for dense layers,
> > > * The breadth and consistency of empirical results across architectures and domains, including structured comparisons against TRL/TCL/TT in both CNNs and ViTs,
> > > * The clear inductive-bias narrative (including limitations) and the theoretical grounding via standard results,
> > > * And the fact that the main concerns are about clarity and framing rather than flawed methodology,
> > >
> > > we respectfully ask you to lean toward acceptance.
> > >
> > > ---
> > >
> > > Our response is split across 4 sequential comments. This is part 4 / 4.

---

### Meta-Review · Area_Chair_idwh · 2026-01-01

**Summary:**

The submission proposes NdLinear, a tensor-native replacement for flattened linear layers via sequential mode-wise transforms, and the reviewers largely agree the idea is intuitive and potentially useful, with impressive parameter/FLOP reductions and broad empirical coverage across many architectures.

However, several core issues remain unresolved in the submitted paper. In particular, (i) the theory and expressivity claims are not stated with the precision and rigor expected, where multiple reviewers found theorems/assumptions unclear or overly suggestive of guarantees that are not actually provided. Addressing this issue would need a significant revision of the paper. (ii) the positioning vs. prior structured linear operators is still not fully convincing across settings, with requests for stronger/centralized comparisons and clearer guidance on when NdLinear is preferable versus alternative structured matrices (and in entangled regimes the evidence remains limited); and (iii) the practical efficiency at scale is not yet substantiated—runtime/memory results are mainly for small/mid-sized experiments, while large-model throughput and distributed-training behavior are left as future work rather than demonstrated.

While the rebuttal offers helpful clarifications and promises concrete improvements (centralized configuration tables, revised figures, clearer scoping of claims, and additional structured baselines in some cases), these changes require significant revision of the current paper. Given these considerations, the paper is better suited for resubmission after consolidating the improvements into the main text.

**Reviewer Concerns:**

*Theory/positioning mostly addressed; rigor still partly outstanding

The rebuttal clarifies the interpretation of NdLinear, tightens the scope of the theory to standard VC/complexity, and explains the novelty in relation to Si et al. However, a more rigorous characterization of function-class/expressivity (beyond conceptual discussion) remains incomplete.

* Experimental clarity improved; some evidence gaps remain

The revision partially addressed the reviewers' concerns with concrete planned edits for the centralized configuration summary, a clearer baseline-tuning description, and improved “Dial-a-Bias” figures and statistics. But higher-dimensional (4D/5D) demonstrations and additional mechanistic validations (e.g., rank/bias propagation diagnostics) are still missing.

* Practical efficiency at scale and broader baselines remain outstanding

While the rebuttal partially address this concern, the key unresolved issues are: (i) large-scale (LLM/distributed) runtime/memory, and (ii) broader structured-baseline coverage (e.g., MONARCH/Kronecker/continuous structured bases) across representative settings

**Reviewer Scores:**

Most reviewers would keep the scores unchanged

---

### Decision · Program_Chairs · 2026-01-26

Reject